https://doi.org/10.1038/s41467-021-27021-0　　**OPEN**

# Cooperative herbivory between two important pests of rice

Qingsong Liu [1,2,5], Xiaoyun Hu[1,5], Shuangli Su[1,5], Yuese Ning[1], Yufa Peng[1], Gongyin Ye [3], Yonggen Lou [3], Ted C. J. Turlings [4] & Yunhe Li [1✉]

Normally, when different species of herbivorous arthropods feed on the same plant this leads to fitness-reducing competition. We found this to be different for two of Asia's most destructive rice pests, the brown planthopper and the rice striped stem borer. Both insects directly and indirectly benefit from jointly attacking the same host plant. Double infestation improved host plant quality, particularly for the stemborer because the planthopper fully suppresses caterpillar-induced production of proteinase inhibitors. It also reduced the risk of egg parasitism, due to diminished parasitoid attraction. Females of both pests have adapted their oviposition behaviour accordingly. Their strong preference for plants infested by the other species even overrides their avoidance of plants already attacked by conspecifics. This cooperation between herbivores is telling of adaptations resulting from the evolution of plant-insect interactions, and points out mechanistic vulnerabilities that can be targeted to control these major pests.

[1] State Key Laboratory for Biology of Plant Diseases and Insect Pests, Institute of Plant Protection, Chinese Academy of Agricultural Sciences, 100193 Beijing, China. [2] College of Life Sciences, Xinyang Normal University, 464000 Xinyang, China. [3] Institute of Insect Sciences, Zhejiang University, 310058 Hangzhou, China. [4] Laboratory of Fundamental and Applied Research in Chemical Ecology, University of Neuchâtel, 2000 Neuchâtel, Switzerland. [5] These authors contributed equally: Qingsong Liu, Xiaoyun Hu, Shuangli Su. ✉email: liyunhe@caas.cn

Species that feed on the same resource are commonly regarded to be competitors[1–3]. In the great majority of cases, species that share a food resource negatively affect each other's performance, and the conventional interspecific competition theory is widely recognized for a diverse range of taxonomic groups including plants, birds, reptiles, marine invertebrates, insects, and microbes[1,4–6]. Yet, there are instances of mutually beneficial interactions between species that feed on the same food source. This is mostly known for microbes that can assist other organisms in various ways to help reach, convert, and digest food[7–9]. In very rare cases, certain vertebrates have also evolved cooperative interactions to better exploit and share resources, as, for instance, has been shown for predators with complementary hunting tactics[10–12].

Amongst arthropods, mutually beneficial interactions have been reported mainly for social insects with food providing partners[13]. The classic example is the symbiotic relationship between ants and aphids, in which aphids produce sugar-rich honeydew that is collected by the ants, and in exchange ants care for and protect the aphids from predators and parasitoids[14,15]. In none of these associations the arthropods share the same food sources, and we are not aware of any example of mutually beneficial exploitation of the same resources by arthropods.

The most common food sources shared among arthropods are plants, with virtually all plants being attacked by a number of different species. This has so far been assumed to always lead to competition[1,3]. The extent of this competition can vary and the consequences can be highly asymmetrical, but in all reported examples the effects of feeding on the same plant are negative for at least one of the herbivores, independently of whether they are phylogenetically close, have the same mode of feeding, or feed on the same tissues[1,16–20]. In certain cases, one herbivore species can benefit from the presence of another herbivore species; for instance, by causing physiological changes in the plants that make these plants less toxic and/or more nutritious[18,19,21–24] or by masking the volatile foraging cues used by natural enemies[25–29]. These types of plant-mediated benefits, which are often referred to as facilitation[30], are frequently studied in the context of interactions between root and shoot herbivores[31–34] and, with a rare exception[35], described as unidirectional, with just one of the species benefitting the other. To our knowledge, there are no known examples of two species of herbivores both seeking and benefitting from simultaneous presence on the same host plants.

It has been postulated that mutually beneficial interactions among species of insect herbivores must exist, but no specific examples have been uncovered yet[22,36]. It is one thing to demonstrate that two herbivores benefit from jointly feeding on the same plant, it is another to conclude that they actively pursue the plant-mediated benefits. Here we propose such a partnership between the brown planthopper (BPH), *Nilaparvata lugens* and the rice striped stem borer (SSB), *Chilo suppressalis*, two of the most devastating pests of rice[37]. Our hypothesis that both insects benefit from attacking the same plant was prompted by our recent finding[27] that BPH performs better on SSB-infested rice plants due to an increased amino acid content, which is beneficial to BPH, and a decrease in levels of toxic beta-sitosterol and campesterol that have been reported to inhibit BPH development[38,39]. We also found that BPH escapes parasitism of its eggs by preferentially ovipositing on rice plants that are already infested by SSB[27]. The apparent reason for this oviposition strategy is that *Anagrus nilaparvatae*, the most common egg parasitoid of BPH, uses volatiles emitted by BPH-infested plants to locate plants with eggs. Plants that are co-infested by SSB release a different blend of volatiles that is not attractive to the parasitoid[27]. Previous work also indicates that SSB larvae perform poorly on rice plants that are already infested by conspecifics due

to induced plant resistance, and that females show a strong oviposition preference for uninfested rice plants relative to SSB-infested rice plants, in accordance with the 'mother knows best' principle[40]. BPH has been shown to suppress certain defenses in rice[41,42]. This raises the question whether sharing host plants with BPH can help SSB to counter the direct defenses of rice plants and possibly mitigate the defense responses to SSB infestation, and, if so, whether it also has adapted its oviposition behavior accordingly.

To answer these questions, we tested the performance of SSB larvae on uninfested rice plants, and on plants infested either by BPH only, by SSB only, or by both species. In cage experiments we tested if the oviposition preference of SSB moths matched the measured performance of their larvae on the differently pre-infested plants. The performance results prompted us to further investigate the molecular and biochemical mechanisms that may be involved, with a focus on the induction and suppression of protease inhibitors in the rice plants. We also examined the effects of BPH infestation on the emission of SSB-induced indirect defense volatiles that attract an important egg parasitoid of SSB. In olfactometer assays and additional cage experiments it was found that the presence of BPH also reduces the risk of SSB eggs to be parasitized. These results combined with the data shown in our previous study[27] well support our hypothesis of a cooperative interaction between two insect herbivores when jointly feeding on the same rice plant. The elucidation of the plant-mediated mechanisms underlying this mutually beneficial cooperation can be the basis for plant breeding strategies to control these two important rice pests.

## Results

**Performance of SSB caterpillars on herbivore-infested rice plants**. When *C. suppressalis* larvae were allowed to feed for 7 days on rice plants that were either uninfested (control), infested by SSB larvae alone (SSB), BPH nymphs alone (BPH), or both SSB larvae and BPH nymphs (SSB/BPH), the body weight of SSB caterpillars was significantly different among the treatments ($F_{3,165} = 8.46$, $P < 0.001$) (Fig. 1a). The body weight of SSB larvae was significantly lower when fed on plants that had been infested by SSB larvae than on all other plant treatments (all $P < 0.01$). Importantly, there was no difference in larval weight among the other three treatments (uninfested plants, feeding on BPH-infested plants or SSB/BPH-infested plants; $P > 0.05$). These results imply that additional infestation by BPH fully eliminated the negative effects of SSB infestation on successively feeding conspecifics. And this effect seems to be independent of infestation sequence, as weight gain was only marginally different between SSB larvae that were placed on dual-infested plants either before or after BPH infestation ($P = 0.051$) (Fig. 1b).

**Oviposition preference of SSB females in greenhouse**. To investigate the oviposition preferences of SSB females, we conducted bioassays under greenhouse as well as field conditions (Figs. 2a and 3). When given a choice between uninfested and SSB-infested plants, SSB females laid far fewer eggs on SSB-infested plants than on uninfested plants (likelihood radio test (LR test) with generalized lineal mixed model (GLMM), $\chi^2 = 25.94$, $df = 1$, $P < 0.001$) (Fig. 2b). However, compared to uninfested plants, the females preferred to lay eggs on BPH-infested plants (LR test with GLMM, $\chi^2 = 7.33$, $df = 1$, $P = 0.007$) (Fig. 2c) or on SSB/BPH-infested plants (LR test with GLMM, $\chi^2 = 4.59$, $df = 1$, $P = 0.03$) (Fig. 2d). As expected, SSB females also laid significantly more eggs on BPH-infested or SSB/BPH-infested plants relative to SSB-infested plants as shown in Fig. 2e, f (LR test with GLMM, $\chi^2 = 7.33$, $df = 1$, $P < 0.001$ for BPH versus

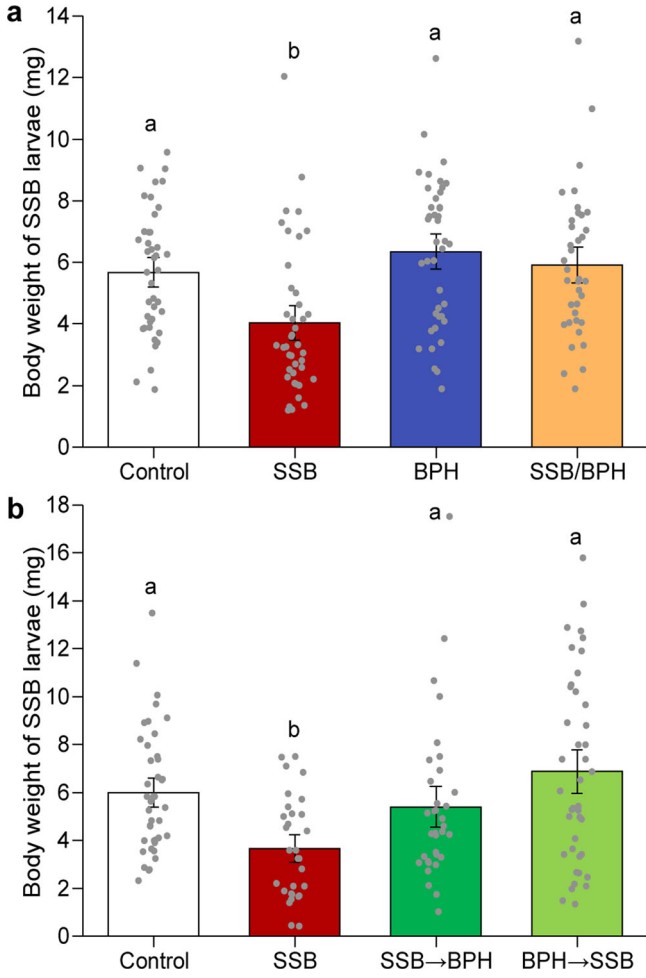

**Fig. 1 Weight of SSB larvae after 7 days of feeding on differentially infested rice plants. a** Neonates of rice striped stem borer (SSB) were individually placed on rice plants that were either uninfested (Control) ($n = 41$), infested by SSB larvae only (SSB) ($n = 46$), brown planthopper (BPH) only (BPH) ($n = 41$), or both SSB and BPH (SSB/BPH, the two species were simultaneously introduced to the plants) ($n = 37$). Exact $P$-values = 0.0012 (Control vs SSB), 0.16 (Control vs BPH), 0.64 (Control vs SSB/BPH), 4.85e-06 (SSB vs BPH), 3.20e-04 (SSB vs SSB/BPH), and 0.37 (BPH vs SSB/BPH). **b** Neonates of SSB were individually placed on rice plants that were either uninfested (Control) ($n = 42$), infested by SSB only (SSB) ($n = 30$), or both SSB and BPH in sequencing order (SSB → BPH ($n = 34$), plants were infested with SSB larvae for 24 h then BPH were added for another 24 h; BPH → SSB ($n = 43$), plants were infested with BPH for 24 h then SSB larvae were added for another 24 h). Exact $P$-values = 0.0023 (Control vs SSB), 0.41 (Control vs SSB → BPH), 0.23 (Control vs BPH → SSB), 0.029 (SSB vs SSB → BPH), 4.51e-05 (SSB vs BPH → SSB), and 0.051 (SSB → BPH vs BPH → SSB). Bars indicate mean ± SE. Data were analyzed using two-way analysis of variance followed by least significant difference (LSD) post-hoc test. Different letters above the bars indicate significant differences between treatments ($P < 0.05$).

SSB; $\chi^2 = 112.77$, $df = 1$, $P < 0.001$ for SSB/BPH versus SSB), whereas they laid similar numbers of eggs on BPH-infested and SSB/BPH-infested plants (LR test with GLMM, $\chi^2 = 1.66$, $df = 1$, $P = 0.20$) (Fig. 2g).

When SSB females were offered the four types of rice plants simultaneously, SSB females laid significantly different numbers of eggs among treatments (Fig. 2h; LR test with GLMM, $\chi^2 = 1371.14$, $df = 3$, $P < 0.001$), with the most eggs on plants infested by both SSB and BPH, and slightly less on BPH-infested

plants and uninfested plants. Importantly, SSB females laid only very few eggs on SSB-infested plants, significantly less than on the other three treatments (all $P < 0.001$) (Fig. 2h).

**Oviposition preferences of SSB females under field conditions.** Consistent with the results from the greenhouse experiments, far more eggs were laid on uninfested plants than on SSB-infested plants in field cage experiments (LR test with GLMM, $\chi^2 = 52.08$, $df = 1$, $P < 0.001$) (Fig. 3a). Compared to uninfested rice plants, the females preferred to lay eggs on BPH-infested plants (LR test with GLMM, $\chi^2 = 26.34$, $df = 1$, $P < 0.001$) or SSB/BPH-infested plants (LR test with GLMM, $\chi^2 = 4.61$, $df = 1$, $P = 0.03$) (Fig. 3b, c). When given a choice between SSB-infested plants and SSB/BPH-infested plants, the females preferred to lay eggs on the co-infested plants, as expected (LR test with GLMM, $\chi^2 = 26.42$, $df = 1$, $P < 0.001$) (Fig. 3d).

**Gene expression changes in rice plants.** RNA-seq analysis was carried out to assess gene expression changes in response to infestation by SSB, BPH or both species for 48 h. A partial least squares-discriminant analysis (PLS-DA) on all 12 transcriptomic datasets indicated there was a significant difference in total gene expression across the four treatments (PLS-DA, classification error = 44.2%, $P = 0.04$) (Fig. 4a). As shown in Fig. 4a, the first two principal components explained 39.5% (PLS1), and 13.3% (PLS2) of the total variance, respectively. PLS1 revealed a clear separation of samples with SSB infestation (SSB and SSB/BPH) from others (BPH and control). And PLS2 separated samples with BPH infestation (BPH and SSB/BPH) from others (SSB and control). Taken together, these results suggest that each infestation regime had distinctly different effects on the gene expression of rice plants.

The gene expression analyses further showed that feeding by SSB resulted in the differential expression of 12512 genes (absolute $\log_2$(fold change) > 0 and $P < 0.05$), of which 6533 genes were upregulated and 5979 genes downregulated. Infestation by BPH alone induced a relative weak response of rice plant, that is, 2523 differentially expressed genes including 1292 upregulated genes and 1231 downregulated genes. Co-infestation by the two species induced the upregulation of 3640 genes and the downregulation of 4082 genes (Fig. 4b). Interestingly, compared to SSB-infested plants, 992 genes were downregulated when plants were co-infested with BPH and SSB, and the gene ontology (GO) enrichment analysis showed that these repressed genes were largely associated with defense responses, such as jasmonic acid-related processes, and enzyme inhibitor activity (Supplementary Data 1).

**Jasmonic acid and salicylic acid associated gene expression and their accumulation.** A total of 10392 *Arabidopsis* orthologs of rice genes were included in phytohormone-related gene expression signatures (Supplementary Data 2). The results showed that, when compared to uninfested plants, genes associated with jasmonic acid (JA) and salicylic acid (SA) pathways were generally upregulated in plants infested by SSB larvae, BPH nymphs, or both species (Fig. 4c). However, compared to SSB-infested plants, dual infestation by SSB and BPH resulted in an apparent downregulation of both JA and SA pathways in rice (Fig. 4c). The gene expression of other common plant hormones, including ethylene, cytokinins, abscisic acid, gibberellins, and auxin, did not show consistent correlations with different pest infestations and was not included in further analyses (Supplementary Fig. 1). Four genes involved in JA biosynthesis, *OsLOX9*, *OsJAR1;2*, *OsDAD1;3*, and *OsAOC*, were significantly upregulated upon SSB infestation, but were not induced when plants were co-infested by both BPH

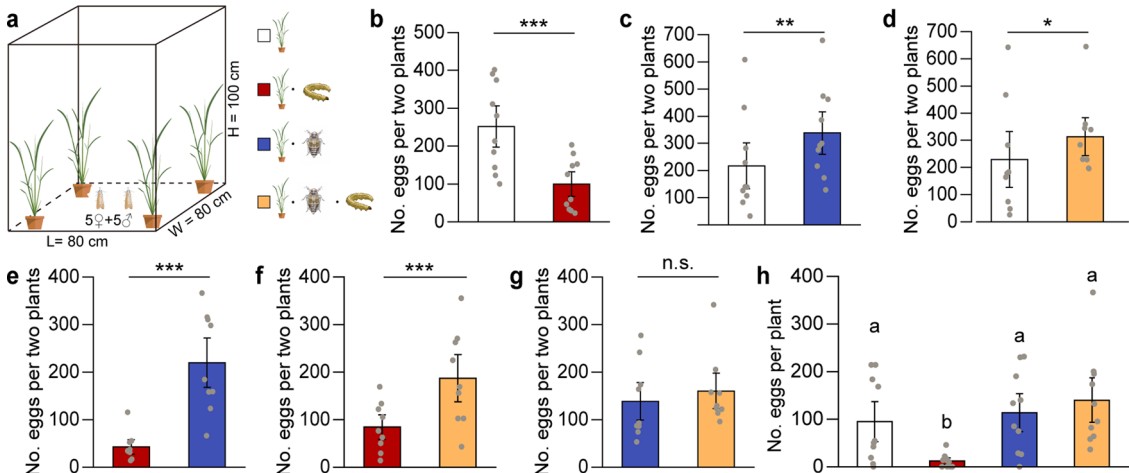

**Fig. 2 Oviposition preference of SSB moths in greenhouse experiments.** Number of eggs laid by female *C. suppressalis* on rice plants that were either uninfested (Control), infested by SSB larvae only (SSB), BPH only (BPH), or by both SSB and BPH (SSB/BPH). **a** Schematic drawing of the oviposition experiments. **b** Control versus SSB larva-infested plants ($n = 10$). **c** Control versus BPH-infested plants ($n = 11$). **d** Control versus dually infested plants ($n = 9$). **e** SSB larva-infested plants versus BPH-infested plants ($n = 9$). **f** SSB larva-infested plants versus dually infested plants ($n = 9$). **g** BPH-infested plants versus dually infested plants ($n = 9$). **h** SSB female moths were exposed to the four types of rice plants all together ($n = 10$). For **b**–**g**, exact *P*-values = 3.52e-07 (Control vs SSB), 0.0068 (Control vs BPH), 0.032 (Control vs SSB/BPH), <2.2e-16 (SSB vs BPH), 9.03e-05 (SSB vs SSB/BPH), 0.20 (BPH vs SSB/BPH); For **h**, exact *P*-values = 2.15e-04 (Control vs SSB), 0.87 (Control vs BPH), 0.22 (Control vs SSB/BPH), <1e-04 (SSB vs BPH), <1e-04 (SSB vs SSB/BPH), and 0.67 (BPH vs SSB/BPH). Two-sided likelihood ratio test with generalized linear mixed model was conducted for the number of eggs (Poisson distribution error). Each bar represents the mean ± SE. Data columns with asterisks (\*\*\**P* < 0.001, \*\**P* < 0.01, \**P* < 0.05, or with different small letters (*P* < 0.05) indicate significant differences between treatments; n.s. indicates a nonsignificant difference (*P* > 0.05).

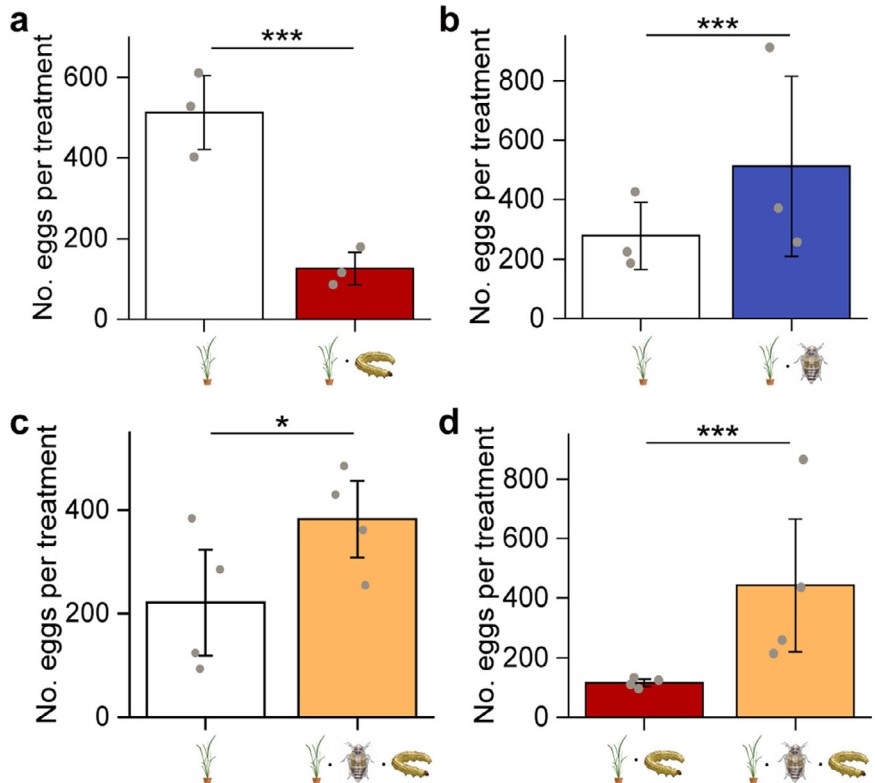

**Fig. 3 Oviposition preference of SSB moths in field cage experiments.** Number of eggs laid by female *C. suppressalis* on rice plants that were either uninfested (Control), infested by SSB larvae only (SSB), BPH only (BPH), or both SSB and BPH (SSB/BPH). **a** Control versus SSB larva-infested plants ($n = 3$). **b** Control versus BPH-infested plants ($n = 3$). **c** Control versus dually infested plants ($n = 4$). **d** SSB larva-infested plants versus dually infested plants ($n = 4$). Exact *P*-values = 5.30e-13 (Control vs SSB), 2.86e-07 (Control vs BPH), 0.032 (Control vs SSB/BPH), and 2.74e-07 (SSB vs SSB/BPH). Two-sided likelihood ratio test with generalized lineal mixed model was conducted for the number of eggs (Poisson distribution error). Each bar represents the mean ± SE. Data columns with asterisks (\*\*\**P* < 0.001, \**P* < 0.05) indicate significant differences between treatments.

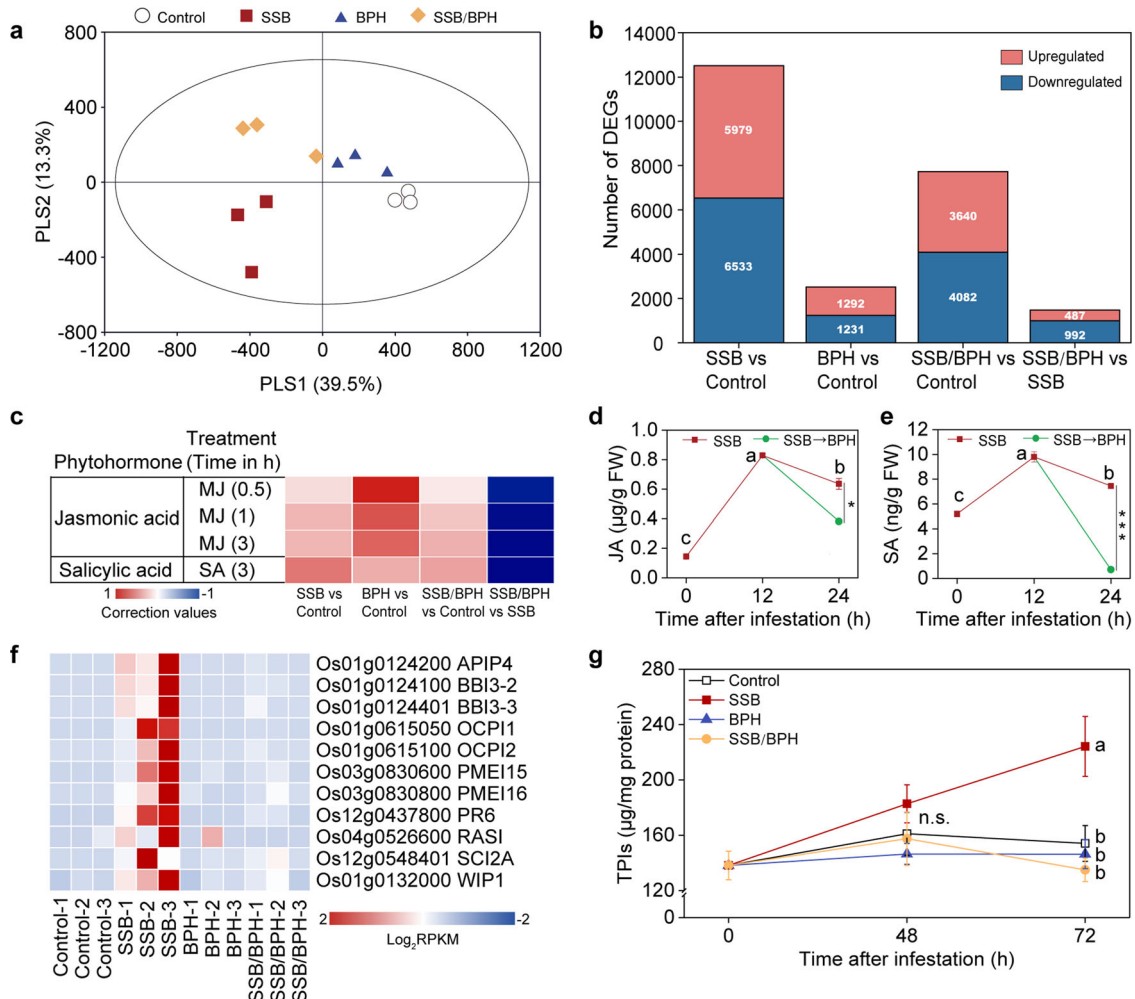

**Fig. 4 Rice plant responses to herbivore infestation. a** Partial least squares-discriminant analysis (PLS-DA) of all detected genes in rice plants that were either untreated (control), infested by SSB, BPH, or both herbivores (SSB + BPH) for 48 h (three biological replicates per treatment). The percentage of variation of the data explained by principal component 1 (PLS1) and PLS2 is in parentheses (39.5 and 13.3%, respectively). The score plot displays the grouping pattern according to the first two components and the ellipse defines the Hotelling's T2 confidence interval (95%) for the observations. **b** Differentially expressed genes among differently treated rice plants. **c** Hormonometer analyses for jasmonic acid (JA) and salicylic acid (SA) signatures based on transcriptomic responses of rice to herbivory. The colors indicate similarity between herbivore infestation and a particular hormone response (blue and red for negative and positive correlations, respectively, see bottom). MJ methyl jasmonate, SA salicylic acid. The contents of endogenous JA (**d**) and SA (**e**) in rice plants subjected to SSB infestation or SSB plus BPH in sequence. Significance was determined by one-way ANOVA followed by Tukey's honest significant difference (HSD) test (for three groups) or two-sided Student's $t$-test (for two groups). For JA contents, exact $P$-values = 1.79e-06 (0 h vs 12 h), 1.18e-05 (0 h vs 24 h), 0.0022 (12 h vs 24 h), 0.020 (SSB 24 h vs SSB/BPH 24 h); for SA contents, exact $P$-values = 8.05e-05 (0 h vs. 12 h), 0.0040 (0 h vs 24 h), 0.0032 (12 h vs 24 h), and 1.68e-06 (SSB 24 h vs SSB/BPH 24 h) ($n = 3$). Different letters indicate significant differences within the SSB treatment at different time points ($P < 0.05$); asterisks indicate significant differences between SSB and SSB → BPH treatments (***$P < 0.001$, *$P < 0.05$) ($n = 3$). Bars indicate mean ± SE. **f** Heatmap of the expression of 11 enzyme inhibitors genes. Log₂-transformed RPKM values are plotted. APIP4 bowman-birk inhibitor AvrPiz-t interacting protein 4, BBI3-2 bowman-birk inhibitor 3-2, BBI3-3 bowman-birk inhibitor 3-3, OCPI1 *Oryza sativa* chymotrypsin inhibitor-like 1, OCPI2 *Oryza sativa* chymotrypsin inhibitor-like 2, PMEI 15 pectin methylesterase inhibitors 15, PMEI 16 pectin methylesterase inhibitors 16, PR6 pathogenesis-related proteins 6, RASI rice alpha-amylase/subtilisin inhibitor, SCI2A subtilisin-chymotrypsin inhibitor-2A, WIP1 wound-induced protease inhibitor 1. FPKM fragments per kilobase of transcript per million fragments mapped. **g** Time course of the contents of trypsin protease inhibitors (TPIs) in rice plants that were either uninfested (Control), infested by SSB, BPH, or both ($n = 3$). For 48 h, exact $P$-values = 0.73 (Control vs SSB), 0.89 (Control vs BPH), 0.10 (Control vs SSB/BPH), 0.36 (SSB vs BPH), 0.63 (SSB vs SSB/BPH), and 0.95 (BPH vs SSB/BPH).; for 72 h, exact $P$-values = 0.034 (Control vs SSB), 0.98 (Control vs BPH), 0.78 (Control vs SSB/BPH), 0.020 (SSB vs BPH), 0.0096 (SSB vs SSB/BPH), and 0.94 (BPH vs SSB/BPH). Bars indicate mean ± SE. Data were analyzed using one-way ANOVA followed by Tukey's post-hoc test. Different letters indicate significant differences between treatments ($P < 0.05$). n.s. not significant ($P > 0.05$).

and SSB (Supplementary Data 3). Notably, five genes involved in JA signaling (*OsLOXL-2*, *OsAOS3*, *OsJAZ1*, *OsJAZ7*, and *OsJAZ9*) and nine SA-responsive genes (*OsPR2*, *OsPR4*, *OsPR4B*, *OsPR4C*, *OsPR4D*, *OsPR6*, *OsPR10*, *OsPR10A*, and *OsPR10B*) were activated by SSB infestation, but were suppressed by dual infestation (Supplementary Data 4).

Because of the important roles of transcription factors (TFs) in plant responses to insect infestation, we further explored if the promoters of the JA pathway genes are enriched in binding sites for certain transcription factors (TFs). Based on the analysis, we predicted that the four TFs, OsNAC39 (*Os03g0327100*), OsIDD1 (C2H2 type family protein) (*Os03g0197800*), OsHOX21 (HD-ZIP

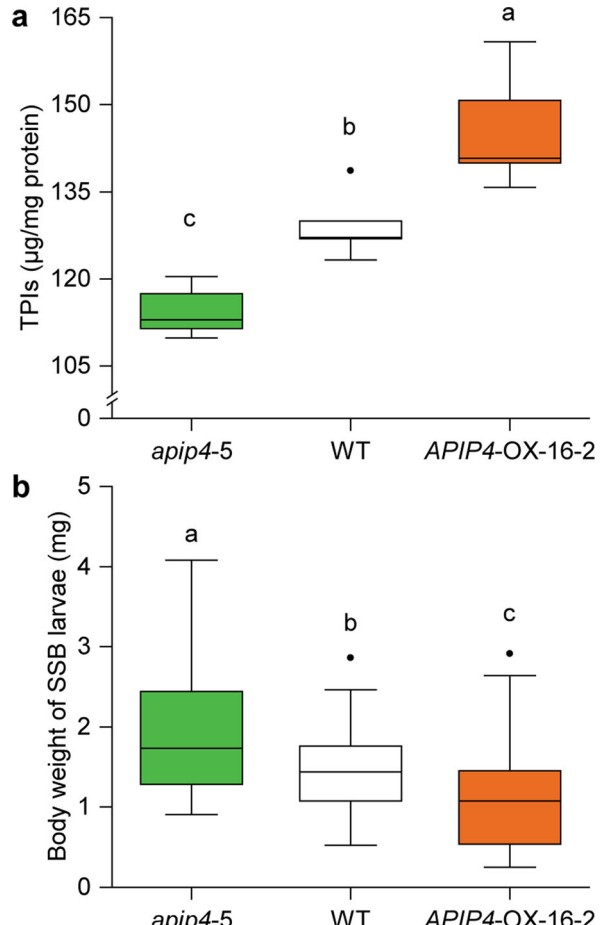

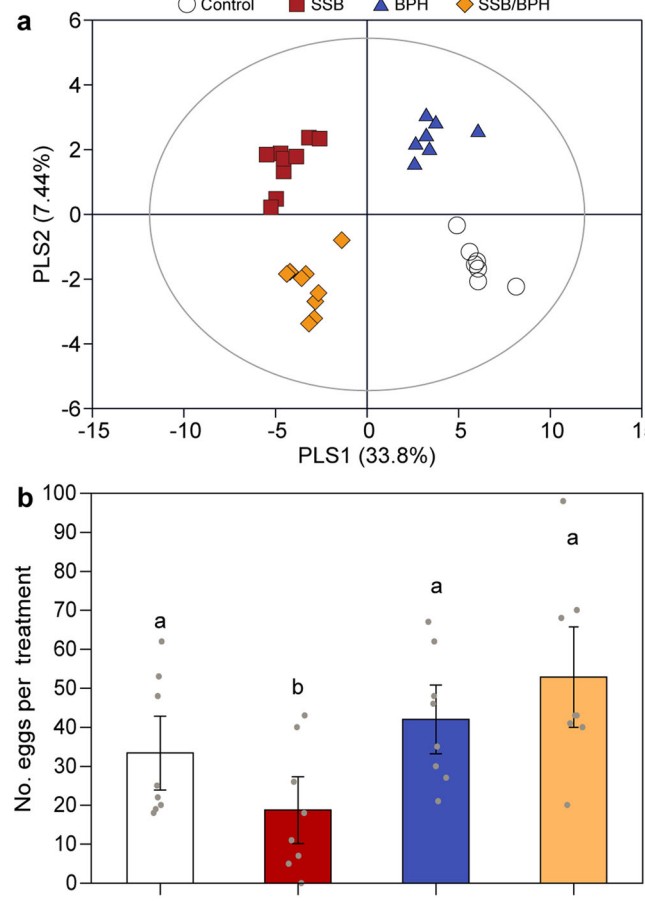

**Fig. 5 Bowman-birk inhibitor *AvrPiz-t interacting protein 4* (*APIP4*) positively regulates TPIs production and SSB resistance. a** The contents of trypsin protease inhibitors (TPIs) in *apip4*-5, *APIP4*-OX-16-2, and wild-type (WT) Nipponbare (NPB) rice plants (*n* = 5). Exact *P*-values = 0.018 (WT vs *apip4*-5), 0.0093 (WT vs *APIP4*-OX-16-2), and 4.90e-05 (*apip4*-5 vs *APIP4*-OX-16-2). **b** Weight of SSB larvae after five days of feeding on *apip4* lines (*n* = 38) and *APIP4*-OX lines (*n* = 53) or WT rice plants (*n* = 36). Exact *P*-values = 5.46e-03 (WT vs *apip4*-5), 4.72e-04 (WT vs *APIP4*-OX-16-2), and 5.21e-10 (*apip4*-5 vs *APIP4*-OX-16-2). Boxplots indicate median (middle line), 25th, 75th percentiles (box), and maximum and minimum values (whiskers) as well as outliers (single points). Different letters indicate significant differences between treatments (*P* < 0.05, one-way analysis of variance followed by Tukey's honest significant difference test).

**Fig. 6 Volatiles released by rice plants and effect on oviposition behavior of *C. suppressalis* females. a** Projection to latent structures-discriminant analysis (PLS-DA) of volatile emissions produced by rice plants that were either untreated (Control), infested by SSB, BPH, or both herbivores (SSB/BPH) for 48 h. The score plot display the grouping pattern according to the first two components and the ellipse defines the Hotelling's T2 confidence interval (95%) for the observations. **b** Number of eggs laid by female *C. suppressalis* near or on filter paper that had been treated with volatiles collected from four types of differently treated rice plants. Exact *P*-values = 0.022 (Control vs SSB), 0.68 (Control vs BPH), 0.20 (Control vs SSB/BPH), <0.001 (SSB vs BPH), <0.001 (SSB vs SSB/BPH), and 0.82 (BPH vs SSB/BPH). Each bar represents the mean ± SE. Significance was determined by two-sided likelihood ratio test with generalized lineal mixed model (Poisson distribution error). Different letters above bars indicate significant differences between treatments (*P* < 0.05; *n* = 8).

family protein) (*Os03g0170600*), and *OsSta2* (ERF family protein) (*Os02g0655200*), bind to promoters of genes associated with the JA pathway (Supplementary Data 5). Moreover, the expression patterns of these TFs and JA pathway genes were similar (Supplementary Data 5), strongly suggesting that these TFs control the expression of the identified JA pathway genes.

Consistent with the phytohormone-related gene expression results, SSB infestation induced a significant increase in the levels of both JA (12 h vs 0 h, *P* < 0.001; 24 h vs 0 h, *P* < 0.001) and SA (12 h vs 0 h, *P* < 0.001; 24 h vs 0 h, *P* = 0.004; Fig. 4d, e). However, when BPH nymphs were added after 12 h infestation by SSB alone, the levels of JA and SA in rice plants significantly decreased (JA: *P* = 0.02; SA: *P* < 0.001; Fig. 4d, e).

**Protease inhibitor-associated gene expression and their accumulation.** When determining GO terms that were significantly enriched (Benjamini–Hochberg false discovery rate adjusted *P*-value < 0.05) in the set of 992 downregulated differentially expressed genes (DEGs) between dual infestation samples and SSB-infestation samples, we identified several molecular function terms that were associated with protease inhibitor activity including serine-type endopeptidase inhibitor activity (GO:0004867), enzyme inhibitor activity (GO:0004857), endopeptidase inhibitor activity (GO:0004866), and peptidase inhibitor activity (GO:0030414) (Supplementary Data 1). After screening for genes involved in these categories, we found 11 protease inhibitor genes that were strongly induced by SSB infestation, but were significantly attenuated by the additional infestation of BPH nymphs (Fig. 4f). The expression of nine genes was validated by qRT-PCR, showing similar expression patterns

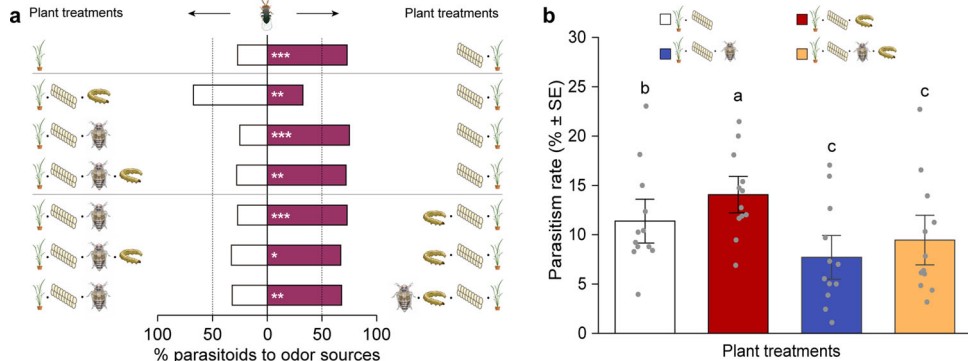

**Fig. 7 Preferences of females of the egg parasioid *Trichogramma japonicum*. a** Choice of *T. japonicum* wasps when offered the odor of differently treated plants in a Y-tube olfactometer. Bars represent the percentages of wasps choosing either of the odor sources. Asterisks indicate significant differences from a 50:50 distribution (binomial test: \*$P < 0.05$, \*\*$P < 0.01$, \*\*\*$P < 0.001$; $n = 58$–$80$) Exact $P$-values and number of replicates were as following: $P = 5.84e$-$04$, $n = 59$ (plants with eggs vs uninfested plants); $P = 2.32e$-$03$, $n = 80$ (plants with SSB and eggs vs plants with eggs); $P = 6.74e$-$05$, $n = 67$ (plants with eggs vs plants with BPH and eggs); $P = 1.07e$-$03$, $n = 60$ (plants with eggs vs plants with SSB, BPH, and eggs); $P = 1.42e$-$04$, $n = 73$ (plants with SSB and eggs vs plants with BPH and eggs); $P = 0.012$, $n = 58$ (plants with SSB and eggs vs plants with SSB, BPH, and eggs); $P = 8.64e$-$03$, $n = 59$ (plants with SSB, BPH, and eggs vs plants with BPH and eggs). **b** Parasitism rates of SSB eggs by *T. japonicum* in the greenhouse experiments. Exact $P$-values = $4.47e$-$04$ (Control vs SSB), $2.13e$-$08$ (Control vs BPH), $2.79e$-$11$ (Control vs SSB/BPH), <$1e$-$04$ (SSB vs BPH), <$1e$-$04$ (SSB vs SSB/BPH), and $0.14$ (BPH vs SSB/BPH). Each bar represents the mean ± SE. Significance was determined by two-sided likelihood ratio test with generalized lineal model (binomial distribution error). Different letters above bars indicate significant differences between treatments ($P < 0.05$; $n = 12$).

among the four treatments as obtained with RNA-seq (Supplementary Fig. 2), confirming the reliability of the RNA-seq data.

Prompted by the observed changes in protease inhibitor-associated gene expression, we further measured the trypsin protease inhibitors (TPIs) content in rice plants responding to different herbivore infestations. The results showed a significant increase in TPIs content after 72 h SSB infestation compared to uninfested plants ($P = 0.03$). This TPIs content increase was not observed for BPH-infested plants or SSB/BPH-infested plants (Fig. 4g).

**Performance of *C. suppressalis* larvae on TPI mutant and overexpression lines.** We found significant differences in the contents of TPIs among the knockout (*apip4*-5) and overexpression (*APIP4*-OX-16-2) mutants of the bowman-birk inhibitor *AvrPiz-t interacting protein 4* (*APIP4*) as well as WT rice plants ($F_{2,12} = 23.47$, $P < 0.01$) (Fig. 5a). As expected, TPIs content was significantly higher in *apip4*-5 plants ($P < 0.05$) and significantly lower in *APIP4*-OX-16-2 lines ($P < 0.01$) relative to WT plants. Consistent with TPIs contents, the mean body weight of SSB caterpillars was significantly greater (1.33 fold) when fed on *apip4*-5 plants than on WT plants ($P < 0.01$) (Fig. 5b). In contrast, the larval weight of SSB caterpillars fed on *APIP4*-OX-16-2 plants was significantly decreased (by 28.03%; $P < 0.001$), compared to those fed on WT plants.

**Volatile profiles of rice plants and their effects on the oviposition behavior of SSB females.** A total of 61 compounds were detected in the headspace of the four plant treatments (Supplementary Data 6). Plants infested by both SSB and BPH emitted the highest amounts of volatiles, followed by plants infested by SSB, plants infested by BPH, whereas control plants emitted the lowest amounts of volatiles. A PLS-DA using the contents of all detected volatiles revealed significant differences among the four treatments (PLS-DA, classification error = 19.1%, $P = 0.001$) (Fig. 6a). The first two significant components explained 33.8% and 7.44% of the total variance, respectively. Consistent with gene expression data, the first component showed a clear separation between plant infested with SSB (SSB and SSB/BPH) from others

(BPH and Control), and the second component separated BPH-infested samples and SSB-infested samples (BPH and SSB) from others (SSB/BPH and Control).

Using volatiles that had been collected from plants subjected to the four types of treatments as odor sources (applied on filter paper strips), we tested the oviposition preference of SSB females. They differently distributed their eggs among the four treatments (Fig. 6b, LR test with GLMM, $\chi^2 = 26.39$, $df = 3$, $P < 0.001$), with the highest number of eggs observed on or near filter paper that had been treated with volatiles collected from SSB/BPH-infested plants, which was statistically similar to the numbers of eggs laid near or on filter paper with volatiles from BPH-infested plants and uninfested plants. Importantly, SSB females laid significantly lower numbers of eggs on filter paper treated with volatiles collected from SSB-infested plants compared to any of the other treatments (all $P < 0.05$) (Fig. 6b).

**Responses of *T. japonicum* wasps to herbivore-infested rice plants.** In a dual-choice assay, the *T. japonicum* wasps showed a strong preference for plants infested by SSB eggs over uninfested intact plants (binomial test, $P = 0.004$) (Fig. 7a). When offered plants carrying SSB eggs in both sides of the olfactometer, the wasps exhibited a significant preference for plants that were additionally infested by SSB larvae relative to plants infested by SSB eggs alone ($P = 0.002$). In contrast, the wasps were significantly more attracted to plants with eggs alone than plants that were additionally infested by BPH nymphs ($P < 0.001$) or both SSB larvae and BPH nymphs ($P = 0.001$). We further tested the preferences of wasps that were offered plants with combinations of herbivore and egg infestation. The results (Fig. 7a) showed that *T. japonicum* females significantly preferred rice plants with SSB and eggs over plants co-infested by BPH and eggs ($P < 0.001$) or plants with SSB, BPH, and eggs ($P = 0.01$). Finally, as expected, the wasps showed a significant preference for the odor of plants with the combination of SSB, BPH, and eggs over the odor of plants with BPH and eggs ($P = 0.009$). These results imply that additional infestation by BPH changes rice volatiles so that *T. japonicum* wasps no longer recognize and locate the rice plants carrying SSB eggs.

**Parasitism rates of SSB eggs by *T. japonicum* wasps**. In the greenhouse cages with the four plant treatments, the rate of parasitism of SSB eggs by *T. japonicum* wasps differed significantly (LR test with generalized lineal model (GLM), $\chi^2 = 138.178$, $df = 3$, $P < 0.001$). The parasitism rate was highest on plants infested with SSB larvae only (Fig. 7b), which was significantly higher than on uninfested plants or plants infested with BPH only or plants infested with both herbivore species (all $P < 0.001$). The lowest parasitism rate of SSB eggs was observed on plants infested by BPH only. The parasitism was also significantly lower on plants infested by both species compared to control plants ($P < 0.001$). The parasitism rates of SSB eggs in the cages nicely reflected the trends in responses of the parasitoids in the olfactometer (Fig. 7b).

## Discussion

Defining *cooperation* as "any interaction in which an actor confers a fitness benefit to another individual and receives an (inclusive) fitness benefit in return"[43], we may conclude that the observed mutually beneficial plant-mediated interactions between SSB and BPH indeed represent an example of cooperation between two herbivores. We found that not only do the herbivores directly (mitigated plant toxicity) and indirectly (reduced exposure to parasitoids) benefit from jointly attacking plants, we also found that both have adapted their host plant selection and oviposition behavior to optimize the benefits that they derive from each other.

Our previous work shows that BPH prefers to feed and lay eggs on SSB-infested rice plants, which are more nutritious and on which their eggs escape parasitism[27,38]. We confirmed that BPH performs better when fed SSB-infested rice plants with an additional bioassay (Supplementary Fig. 3). The current study demonstrates similar, even stronger benefits for SSB when infesting plants that are already under attack by BPH. The feeding assays show that SSB infestation induces rice defense responses that cause significantly reduced fitness in SSB larvae that subsequently feed on the same plants (Fig. 1). Remarkably, additional infestation by BPH completely neutralized the negative effects of SSB defense induction; SSB caterpillars placed on plants already infested by conspecifics grew considerably better on plants that were also attacked by BPH, independent of order of infestation. On BPH-infested plants the caterpillars performed just as well as on previously uninfested control plant (Fig. 1)

RNA-seq and biochemical analyses showed that infestation by BPH suppresses a broad spectrum of defense-related genes (Fig. 4f, Supplementary Data 1), with one of the main consequences being a significant reduction in SSB-induced production of proteinase inhibitors (Fig. 4g). These are key defensive compounds of rice plants that are particularly effective against chewing herbivores, including SSB (Fig. 5)[44,45]. We also found that co-infestation with BPH suppresses the expression of SA- and JA-associated genes that are normally upregulated by SSB infestation (Fig. 4c, Supplementary Data 4). This suppression led to reduced levels of JA and SA in the plants (Fig. 4d, e). The JA signal transduction pathway is responsible for the production of TPIs in rice plant, and it is known that SSB performs better on JA-deficient mutant rice lines mainly due to reduced TPIs levels[45,46]. Collectively, this insight explains why BPH feeding neutralizes the negative effects of SSB defense induction.

Insect attack typically induces defense responses in plants[47–50], but it is also increasingly evident that various insect herbivores have the ability to, at least partially, suppress plant defenses by interfering with JA or SA biosynthesis and that this can enhance their own performance and that of conspecifics[38,51–54]. This can also benefit certain other species that feed on the same plants. For example, the silverleaf whitefly (*Bemisia tabaci*) activates SA signaling and represses JA-regulated defenses, leading to enhanced nymphal development of this insect[36], which also benefits spider mites[29]. None of these studies appear to show reciprocal benefits for species feeding on the same plant. Here we found that additional infestation of BPH on SSB-infested plant suppressed the expression of genes associated with JA and SA pathway and thereby the contents of JA and SA. Based on in-silico analyses we predict that the TFs including *OsNAC39*, *OsIDD1*, *OsHOX21*, and *OsSta2* control the expression of the downregulated JA pathway genes. Further biochemical and genetic analyses are required to unravel this possible function of the TFs in regulating JA pathway genes.

Consistent with 'mother knows best' hypothesis[55], SSB females avoid to lay eggs on rice plants that are already infested by conspecifics, thus ensuring that their offspring evade the negative effects that SSB-induced defenses[40]. Here we show that SSB females have adapted their oviposition behavior to preferentially oviposit on BPH-infested rice plants, independent of whether SSB larvae are already present or not, as compared to healthy plants (Figs. 2 and 3). By doing so, they benefit from the BPH-mediated suppression of rice defense responses (Fig. 1a). However, the performance of SSB larvae was not any better on rice plants that were already co-infested by SSB plus BPH compared to their performance on healthy plants. So why did SSB females prefer to lay eggs on dual-infested plants rather than on healthy plants (Figs. 2d and 3c)? The experiments with the egg parasitoid *T. japonicum* provide a plausible explanation, as they showed that the presence of BPH significantly reduced the attractiveness of rice plants to the wasp (Fig. 7a). The cage experiment confirmed that the presence of BPH decreased the risk for SSB eggs to be parasitized, implying that the oviposition strategy of SSB females is highly adaptive (Fig. 7b).

It appears that certain well-adapted herbivores can also manipulate the emission of volatiles of their host plants[56]. In most such cases, herbivore infestation suppresses certain key volatile compounds and thereby possibly reduces the attractiveness of plants to certain natural enemies[26,29,52,57,58]. Other herbivores can benefit from this as well. Simultaneous feeding by slugs, for instance, suppress caterpillar-induced volatiles in cabbage plants, thereby reducing the attractiveness of the plants to parasitoids[26]. Similarly, when spider mite-infested Lima bean plants are also infested by whiteflies this leads to a reduced emission of the volatile (E)-β-ocimene compared to plants infested by spider mites only, resulting in a reduced attraction of predatory mites[29]. In other cases, double infestation may actually lead to higher quantities of volatiles being emitted, but the blend is altered in a way that it is no longer attractive to parasitoids[28]. This is also the case for our study system. We previously showed that BPH preferentially oviposits on SSB-infested rice plants, thereby avoiding the attraction of the egg parasitoid *A. nilaparvatae*[27]. These various examples confirm that insect herbivores not only can evolve the ability to use volatiles to identify host plants of better nutritional quality, but also plants where their offspring can escape natural enemies[59].

Our combined results, including the additional tests showing that SSB infestation of rice plants significantly increases the performance of BPH nymphs (Supplementary Fig. 3) strongly supported the conclusion of mutually beneficial interaction between SSB and BPH. This goes against the ingrained notion of competition between phytophagous insects that share a common host plant, and how this competition shapes insect assemblages[1,16,19,60–62]. The resource-based competition theory has been challenged before, but the examples involve specific asymmetric beneficial plant-mediated interactions, meaning that only one of the herbivore species benefits from the presence of

another[22,23,63–67]. In these cases, the benefit is never reciprocal, nor do they seem to represent tightly coevolved interactions with specific behavioral adaptations as found for BPH and SSB in the current study and certain vertebrates[10–12].

The interaction between SSB and BPH reported here seems to represent a highly evolved collaboration to cope with and exploit the direct and indirect defense responses of rice plants. During the coevolutionary arms race between herbivores and host plants, both sides evolve multiple resistance mechanisms against one another. The two herbivores have a long sympatric history together[27,38] and we therefore speculate that the cooperative relationship between SSB and BPH may the result of two opposing coevolutionary arms races that in combination benefit both herbivores. Behavioral comparisons with populations where the two herbivores do not cooccur should help to test this hypothesis.

Although the mutually beneficial interaction between the stem borer and planthopper bears no resemblance to any of the known interactions between other herbivore species attacking a same plant[1,16], the reported type of cooperation is unlikely to be unique. We postulate that agricultural pests are especially prone to rapid evolutionary changes that allow such cooperation to emerge. As pest populations build up in vast monocultures their main challenges revolve around coping with plant defenses and avoiding and resisting their specific natural enemies, whereas finding host plants is no longer a challenge. In such scenarios, different insect species will encounter each other frequently. Unlike the cultivated plants, the insects are subject to natural selection and can evolve traits to jointly overcome plant defenses, without the cultivated plants being able to coevolve to resist these traits. The plants are at the mercy of human selection, which is focused at traits that favor yield and nutritional value, often at the cost of reduced resistance against pests and diseases[68,69]. Yet, as we discover and unravel the intricate adaptations in the insects, we can start steering this human selection in favor of potent pest resistance traits. For the specific example uncovered in our study, interfering with the ability of BPH to suppress the biosynthesis of proteinase inhibitors could be a highly effective and sustainable strategy to control two of rice's most common and most harmful pests.

In summary, the current study reveals a highly adaptive, mutually beneficial relationship between rice planthoppers and stem borers that is mediated by opposing rice plant defense responses. The findings represent an example of a cooperative interaction that challenges traditional interspecific competition theory. The two insect species take advantage of the rice defense responses induced by each other in a manner that suggests that together they are the tentative winners in the arms race with rice plants. The results are also illustrative of the complexity and intricate dynamics of the interaction between plants and insects, and challenge the conventional paradigms of interspecific competition. Future work should further unravel more details about the molecular mechanisms underlying the insect-controlled interactions, which might lead to the development of rice varieties that disrupt the cooperative interaction as potential strategy to control the two pests.

## Methods

**Plants and insects**. Rice (*Oryza sativa*) cultivar Minghui63 was used in this study. Rice plants were grown in a greenhouse at 27 ± 3 °C with 75 ± 10% RH (relative humidity) and a photoperiod of 16:8 h L:D (light:dark). The cultivation of rice plants followed the same procedure as described previously[27]. Plants were used for experiments when they were at the tillering stage, which occurred about 44–49 days after sowing.

*C. suppressalis* larvae were reared on an artificial diet as described[70]. Ten percent honey water solution was provided to supply nutrition for the adults. *N. lugens* were maintained on a BPH-susceptible rice variety Taichung Native 1

(TN1)[38]. *T. japonicum* were obtained from Keyun Industry Co., Ltd (Jiyuan, China). Newly emerged adult wasps were maintained in glass tubes (3.5 cm diameter, 20 cm height) and supplied with 10% honey water solution as a food source and were maintained for at least 6 h to ensure free mating, before females were used for the following experiments. All three species were maintained in climatic chambers at 27 ± 1 °C, 75 ± 5% RH, and a photoperiod of 16:8 h L:D.

**Performance of caterpillars on insect-infested rice plants**. Multiple types of rice plants were prepared: (i) uninfested plants, meaning that potted rice plants remained intact without insect infestation; (ii) SSB-infested plants, each potted rice plant was artificially infested with one 3rd instar SSB larva that had been starved for >3 h for 48 h; (iii) BPH-infested plants, each potted rice plant was artificially infested with a mix of fifteen 3rd and 4th instars BPH nymphs for 48 h; (iv) SSB/BPH-infested plants, each potted rice plant was simultaneously infested with one SSB larva and 15 BPH nymphs for 48 h; (v) SSB → BPH-infested plants, each potted rice plant was artificially infested with one SSB larvae alone for the first 24 h, then 15 BPH nymphs were additionally introduced for another 24 h; (vi) BPH → SSB-infested plants, namely each potted rice plant was artificially infested with 15 BPH nymphs for the first 24 h, then one SSB larvae were additionally introduced for another 24 h. Plant treatments were conducted as described in detail in our previous study[27]. During herbivory treatment, the uninfested plants were placed in a separate room to avoid possible volatile-mediated interference. During the subsequent bioassays, both SSB caterpillar and BPH nymphs remained in or on the rice plants.

Two bioassays were conducted to test the performance of *C. suppressalis* larvae feeding on differently treated rice plants. The first bioassay included the plant treatments i, ii, iii, and vi, and the second bioassay included the plant treatments i, ii, v, and vi. Three 2-day-old larvae of *C. suppressalis* were gently introduced onto the middle stem of each rice plant using a soft brush. The infested rice plants were then placed in climatic chambers at 27 ± 1 °C, 75 ± 5% relative humidity, and a photoperiod of 16:8 h L:D. The *C. suppressalis* larvae were retrieved from the rice plants after 7 days feeding, and they were weighed on a precision balance (CPA2250, Sartorius AG, Germany; readability = 0.01 mg). The mean weight of the three caterpillars on each plant was considered as one biological replicate. The experiment was repeated four times using different batches of plants and herbivores, resulting in a total of 30–46 biological replicates for each treatment.

**Oviposition-preferences of *C. suppressalis* females choosing among differently infested rice plants**

*Greenhouse experiment.* In the greenhouse, seven choice tests were conducted with *C. suppressalis* females including (i) SSB-infested plants versus uninfested plants; (ii) BPH-infested plants versus uninfested plants; (iii) SSB/BPH-infested plants versus uninfested plants; (iv) SSB-infested plants versus BPH-infested plants; (v) SSB-infested plants versus SSB/BPH-infested plants; (vi) BPH-infested plants versus SSB/BPH-infested plants; and (vii) the test in which *C. suppressalis* females were exposed to all four types of rice plants. The experiments were performed as described in detail by Jiao et al.[30]. In brief, four potted plants were positioned in the four corners of a cage (80 × 80 × 100 cm) made of 80-mesh nylon nets for each test. For paired comparisons, two potted plants belonging to the same treatment were placed in opposite corners of each age, and in the test with four types of rice plants, each type of plant was positioned in one of the four corners of each cage. Five pairs of freshly emerged moths (less than 1 day) were released in each cage, and a clean Petri dish (9 cm diameter) containing a cotton ball soaked with a 10% honey solution was placed in the center of the cage as food source. After 72 h, the number of individual eggs on each plant were determined. The experiment was conducted in a greenhouse at 27 ± 3 °C, 65 ± 10% RH, and a photoperiod of 16:8 h L:D. Each choice test was repeated with 9–11 times (replicates).

*Field cage experiment.* The oviposition preference of SSB females was further assessed in a field near Langfang City (39.58° N, 116.48° E), China. Four choice tests were conducted: (i) SSB-infested plants versus uninfested plants; (ii) BPH-infested plants versus uninfested plants; (iii) SSB/BPH-infested plants versus uninfested plants; and (iv) SSB/BPH-infested plants versus SSB-infested plants. The treated rice plants were prepared as described above and were transplanted into experimental plots (1.5 × 1.5 m). For each pairwise comparison, six plots of rice plants were covered with a screened cage (8 × 5 × 2.5 m) made of 80-mesh nylon net to prevent moths from entering or escaping. Each of the six plots contained nine rice plants of a particular treatment, with three plots per cage representing the same treatment. Plots were separated by a 1 m buffer and they were alternately distributed in a 3 × 2 grid arrangement in each cage (Supplementary Fig. 4). Approximately 50 mating pairs of newly emerged *C. suppressalis* adults (<24 h) were released into each cage. After 72 h, the number of individual eggs on each plant was determined. The total number of eggs of three plots in each cage was regarded as one replicate, 3–4 replicates were conducted for each pairwise comparisons.

**Rice plant response to herbivore infestation**

*RNA-seq and data analysis.* To explore the molecular mechanisms underlying the rice plant-mediated interaction between BPH and SSB, gene expression changes in

rice response to infestation by SSB, BPH, or both were analyzed by RNA-seq. The rice plants, uninfested (control) or infested, were prepared as described above. After 48 h, the stems of the plants were harvested and frozen in liquid nitrogen. Samples from five individual plants of the same treatment were pooled together as one biological replicate, and three replicates were collected for each treatment.

RNA-seq analyses were performed as described previously[71]. In brief, total RNA was extracted using the TRIzol reagent (Invitrogen, Carlsbad, CA, USA) and treated with RNase-free DNase I (NEB, Ipswich, MA, USA) to remove any genomic DNA. Library preparation and RNA-seq were performed by Novogene (Beijing, China) using an Illumina Hiseq 4000 system, resulting in ~45–55 million raw reads per sample. Raw reads were subjected to quality checking and trimming to remove adapters, poly-N sequences, and low-quality bases (Phred quality score Q < 20). The yield clean data of each sample were aligned to the rice reference genome IRGSP-1.0 (https://rapdb.dna.affrc.go.jp) using HISAT2 (v2.09)[72], and the number of reads mapped to each gene was counted with featureCounts (v1.5.0-p3)[73]. The expression level of each gene was calculated as FPKM (fragments per kilobase of transcript per million fragments mapped) according to an established protocol[74]. Expression differentiation analyses were conducted with the DESeq2 R package (v. 1.18.0)[75]. Genes with absolute value of $\log_2$(fold change) > 0 and $P$-value < 0.05 were defined as DEGs. The enriched functions of DEGs in RNA-seq datasets were annotated with the Gene Ontology (GO) function using the clusterProfiler R package (v4.0.2)[76], and GO terms with Benjamini–Hochberg false discovery rate (FDR) adjusted $P$-value ($Padj$) <0.05 were considered significantly enriched. The transcriptional signatures of hormonal responses of rice plant to herbivory relative to gene expression in *Arabidopsis* induced by diverse phytohormones was analyzed using Hormonometer program[77]. Since TPIs serve as indicators of induced resistance in rice plants, especially against chewing herbivores such as SSB[44,45], the analyses focused on expression profiles of TPIs-related genes among the four plant treatments. The expression of nine selected TPIs genes were validated by quantitative real-time PCR (qRT-PCR) analyses as previously described[78]. qRT-PCR was conducted on a Bio-RadCFX96 Touch Real-time PCR Detection System instrument (Bio-Rad, Hercules, CA, USA) using TransStart® Top Green qPCR SuperMix (TransGen Biotech, Beijing, China). The rice *ubiquitin 5* gene was used as the internal standard to normalize the variations in gene expression. The primers used are listed in Supplementary Table 1.

*Quantification of endogenous jasmonic and salicylic acid.* JA signaling is well established as the core pathway that regulates chemical defenses in rice plant against herbivores, including SSB and BPH[45,47,50,79]. SA is commonly reported to be involved in cross-talk with JA[36,47,50], and we therefore focused on the analysis of these two major plant hormones. Our RNA-seq results suggested that additional infestation by BPH significantly suppressed the expressions of genes related to JA and SA signaling. Both types of genes are highly upregulated in response to SSB infestation[50]. To confirm this, we quantified the JA and SA levels in rice plants with two treatments: (i) rice plants that were infested with one third-instar SSB larva alone for 24 h; (ii) rice plants that were first infested with one third-instar SSB larva for 12 h and then also with 15 BPH female adults for another 12 h. Rice stems were harvested at three time points: 0 h (uninfested control plants), as well as 12 h and 24 h after infestation. For each treatment, stems from five individual plants were harvested and pooled together as one biological replicate, and three replicates were collected for each time point.

Endogenous measurements of JA and SA were performed by the plant hormone platform at the Institute of Genetics and Developmental Biology, Chinese Academy of Sciences as previously described[80].

*TFs regulation prediction.* As the key regulators of transcription, TFs are important in mediating plant responses to herbivory. To gain more insight into the mechanistic links between hormonal regulation and gene expression changes, we further analyzed the potential regulatory interactions between TFs and the five JA signaling genes (Supplementary Data 4) that were found to be upregulated by SSB infestation but were downregulated by dual infestation. Gene IDs of TFs in rice were retrieved from platform PlantRegMap[81] and were then compared to the downregulated DEGs between dual infestation and SSB-infestation treatments to identify potential regulated TFs. To identify upstream regulatory TFs, the 2000-bp sequences upstream of the start codon of the JA signaling genes were extracted from the Phytozome database (https://phytozome-next.jgi.doe.gov/). Then, these sequences were subjected to the PlantRegMap with a threshold $P$-value ≤ 1e-5 for TF binding site prediction. The motifs of TFs that have the potential to bind to the promoters of the JA signaling genes were screened, and a possible interaction was assigned if there were one or more binding sites of a TF on the promoter of a JA signaling gene. The predicted TFs were matched with the identified TFs that were downregulated in dual infestation as compared to SSB infestation, in order to obtain the potentially co-expressed TFs.

*Quantification of TPIs.* We further measured the accumulation of TPIs in rice plants subjected to insect infestation. These experiments were prompted by the RNA-seq results indicating that the upregulation of TPIs-related genes in response to SSB infestation is significantly suppressed after co-infestation with BPH. The same plant treatments were included as used for RNA-seq but with new batches of plants. Stem samples were collected at 48 h and 72 h of insect infestation. Intact rice

plants that served as controls were also sampled at the same time points. Samples from five individual plants were pooled together as one biological replicate, and three replicates were collected for each treatment. All samples were immediately frozen in liquid nitrogen and stored at −80 °C until further analyses.

TPIs contents were determined using enzyme linked immunosorbent assay (ELISA) kits (J&L Biological, Shanghai, China). The stem samples were ground into a fine power in liquid nitrogen using a mortar and pestle, and each sample (0.1 g) was homogenized in 0.01 M Phosphate Buffered Saline (PBS) buffer (pH = 7.4) (Sigma–Aldrich, St. Louis, MO, USA) with a sample-PBS proportion of 1:9 (1 g plant sample/9 ml of PBS). Samples were centrifuged at $4000 \times g$ for 15 min at 4 °C, and the supernatant was collected. The ELISA experiments were performed following the protocols provided with the kits. The optical density values were recorded at 450 nm using a microplate spectrophotometer (PowerWave XS2, BioTek, Winooski, VT, USA). The protein concentrations in plant samples were measured using a bicinchoninic acid (BCA) protein assay kit (Aidlab Biotechnologies Co., Ltd., Beijing, China) according to the manufacturer's instructions. The amount of protease inhibitor was calculated based on a standard curve, and results were expressed as µg protease inhibitor per mg protein.

### Performance of *C. suppressalis* larvae feeding on TPI mutant and over-expression lines.
To confirm that TPIs is the main defense compound in rice plants that provide resistance against SSB caterpillars, we assessed the performance of SSB larvae on rice lines *apip4-5* with *APIP4* gene (Os01g0124200) deletion and *APIP4-OX-16-2* plants with *APIP4* gene overexpression, as well as on their corresponding wild-type (WT) Nipponbare (NPB) rice plants. These rice lines were generated by CRISPR/Cas9 technology (for details to see Zhang et al.[82]). They were used because our RNA-seq analyses suggested that the *APIP4* gene is an important regulator of TPIs production (Fig. 4). TPIs contents were first measured in stems of each rice line. Rice stem samples from two individual plants were pooled together as one biological replicate, and five replicates were collected for each line. The measurement of TPIs in these rice plants were conducted using ELISA as described above. For the bioassay, 2-day-old SSB larvae were individually fed on plants of each rice line. Larval mass was determined after 5 days of feeding, using a precision balance (CPA2250, Sartorius AG, Germany; readability = 0.01 mg). For each treatment, 36–53 insects were tested. This experiment was conducted in a climatic chamber at $27 \pm 1$ °C, $75 \pm 5\%$ relative humidity, and a photoperiod of 16:8 h L:D.

### Effect of insect-induced volatiles on the oviposition behavior of SSB moths
*Collection and analysis of rice plant volatiles.* Individual rice plants were either uninfested or infested with SSB larvae alone, BPH nymphs alone, or both species simultaneously for 48 h using the method described above. The emitted volatiles were trapped using a dynamic headspace collection system in a climate chamber at $27 \pm 3$ °C, $75 \pm 10\%$ RH, and then analyzed and identified as described[27]. To collect the volatiles, two plants were placed into a glass bottle (3142 ml) connected to an air flow. Air was purified through activated charcoal, molecular sieves (5 Å, beads, 8–12 mesh, Sigma–Aldrich), and silica gel Rubin (cobalt-free drying agent, Sigma–Aldrich) before entering the glass bottle. After 30 min, a volatile collection was started by pushing and pulling air out of the glass bottle at a rate of 400 ml min$^{-1}$ through a glass tube (5 mm diameter, 8 cm height) filled with 30 mg Super Q traps (80/100 mesh, ANPEL Laboratory Technologies (Shanghai) lnc, China) for 3 h (21:00–24:00, the time period when SSB lay their eggs). Volatiles collected on the Super Q traps were extracted with 200 µl of methylene chloride, and 500 ng of nonyl acetate in 10 µl of methylene chloride was added to the samples as an internal standard. The extracts were stored at −30 °C until further analyses. For each treatment, collections were repeated 7–9 times.

A gas chromatograph-mass spectrometer (GCMS QP-2010SE, Shimadzu Ltd., Kyoto, Japan) was used to separate, quantify and identify the collected volatiles. A 1 µL volume of each sample was injected into an RTX-5 MS fused silica capillary column (30 m × 0.25 mm ID × 0.25 µm film thickness; Restek Co., Bellefonte, PA, USA). The inlet was operated in split-less injection mode, and the injector, was maintained at 250 °C. Helium was used as the carrier gas with a flow of 1.0 ml min$^{-1}$ in constant flow mode. The gas chromatograph oven was initially set at a temperature of 40 °C for 2 min before being raised by 6 °C a minute until it reached 250 °C, at which it was kept for 2 min. The mass spectrometer was operated in scan mode with a mass range of 33–300 amu at 5.24 scans s$^{-1}$ and the spectra were recorded in electron impact ionization (EI) at 70 eV. The ion source and mass quadrupole were set at 230 and 150 °C, respectively.

The chemicals were first identified by mass spectral matches to library spectra as well as by matching observed retention time with that of available authentic standards. If standards were unavailable, tentative identifications were made based on referenced mass spectra data available from the National Institute of Standards and Technology (NIST) library (Scientific Instrument Services, Inc, Ringoes, NJ, USA) or based on previous studies[27,40]. Relative quantification of compounds was based on its integrated area relative to the internal standard[27,51].

*Odor preferences of SSB females.* The response of SSB females to volatiles released from differently treated rice plants were investigated to better understand the mechanism underlying the moth's oviposition preferences. The total volatiles emitted from uninfested plants, SSB-infested plants, BPH-infested plants and SSB/BPH-infested plants were collected for this experiment. Plant treatments and

volatiles collections were the same as described above but without the addition of the internal standard. The collected volatiles were diluted in paraffin oil (purity 99%; Sigma–Aldrich, St. Louis, MO, USA) at 1:4 (v/v) and were stored at −80 °C before use.

One milliliter of each of the four types of volatile solutions were separately pipetted on the center of a filter paper strip ($4 \times 21$ cm), which were then hung from the four corners of a cage ($45 \times 45 \times 45$ cm) made of 80-mesh nylon net. Five pairs of freshly emerged SSB moths (<24 h) were released in each cage. After 72 h, the number of eggs deposited on the filter paper strips and the surface of the nylon nets near each paper strip were determined. This oviposition choice test was repeated eight times.

**Response of the egg parasitoid *T. japonicum* to herbivore-infested rice plants.** Multiple types of herbivore-infested rice plants were prepared: (i) uninfested plants (control); (ii) SSB-infested plants; (iii) BPH-infested plants; (iv) SSB/BPH-infested plants; (v) plants infested with SSB eggs (referred to as egg-infested plants); (vi) plant infested with SSB larvae and their eggs (referred to as SSB/egg-infested plants); (vii) plants infested with BPH nymphs and SSB eggs (BPH/egg-infested plants); and (viii) plants infested with both SSB larvae, BPH nymphs, and SSB eggs (referred to as SSB/BPH/egg-infested plants). To prepare these treatments, plants were first artificially infested with herbivores for 48 h as described above, then some of them were subjected to SSB eggs deposition. For that, two potted rice plants of the same type were placed in a cage ($45 \times 45 \times 45$ cm) made of 80-mesh nylon nets, then 30 pairs of freshly emerged moths (<24 h) were released in each cage to mate and lay eggs. After 24 h, the plants were removed from the cage and those that carried 200–250 eggs were used as odor sources. During the period of egg deposition and the subsequent olfactometer experiments with the parasitoid, all insects remained in or on the rice plants.

To test the behavioral responses of *T. japonicum* to differently treated rice plants, they were offered the following pairs of odor sources: (i) uninfested plants versus egg-infested plants; (ii) uninfested plants versus SSB-infested plants; (iii) uninfested plants versus BPH-infested plants; (iv) egg-infested plants versus SSB/egg-infested plants; (v) egg-infested plants versus BPH/eggs infested plants; (vi) SSB/egg-infested plants versus BPH/egg-infested plants; (vii) egg-infested plants versus SSB/BPH/egg-infested plants; (viii) SSB/egg-infested plants versus SSB/BPH/egg-infested plants; and (ix) BPH/egg-infested plants versus SSB/BPH/egg-infested plants.

Responses of *T. japonicum* females to these odor sources were investigated in a Y-tube olfactometer as described[27]. Newly emerged adult wasps were maintained in glass tubes (3.5 cm diameter, 20 cm height) for at least 6 h to ensure that they would mate, before females were used for the experiments. Two rice plants of the same treatment were enclosed in a glass bottle and used as one odor source, and each pair of odor sources was replaced after ten parasitic wasps were tested. For each treatment, a total of 64–88 female wasps were tested. The experiments were conducted between 10:00 and 16:00 on several consecutive days.

**Parasitism rates of *C. suppressalis* eggs by *T. japonicum* wasps.** In a cage experiment, we further tested if the differences in parasitoid attraction observed in the olfactometer for the differentially infested plants can result in differences in parasitism rates of SSB eggs under realistic conditions. The following herbivore-treated plants were prepared as described above: SSB eggs on uninfested, SSB-infested, BPH-infested and SSB/BPH-infested plants. The four types of plants were placed in the four corners of a cage ($60 \times 60 \times 60$ cm) made of 80-mesh nylon nets, respectively. Subsequently, 40 pairs of newly emerged wasps (<1 day old) were released into the cage. After 48 h, the rice leaves with SSB eggs were collected, and the total number of SSB eggs on each plant was counted, and their parasitization status was determined under a microscope two days later; the eggs turned black 3 days after being parasitized. The experiment was replicated 12 times. The experiment was performed in a greenhouse at $27 \pm 3$ °C and with $75 \pm 10\%$ RH and a photoperiod of 16:8 h L:D.

**Statistical analyses.** Statistical analyses were conducted using SPSS 22.0 (IBM SPSS, Somers, NY, USA), R (version 4.0.4, https://www.r-project.org), Microsoft Excel 2019, and SIMCA 14.1 software (Umetrics, Umeå, Sweden). All data were checked for normality and equality of variances prior to statistical analysis. Datasets that did not fit assumptions were square-root (sqrt) transformed to meet the requirements of equal variance and normality. Likelihood ratio test (LR test) applied to a generalized lineal mixed model (GLMM) for overdispersion and grouped design were conducted to compare the number of eggs laid by SSB females on rice plants (Poisson error structure with log link function) using R package lme4 (v1.1-27.1). We used cage ID as random factor to account for the nonindependence between observations within the same cage. Whenever the dispersion value was higher than 2 in the Poisson GLMM, overdispersion was accounted for by using an observation level factor as a random factor. Thus, cage ID and an observation level factor were the two random factors in the mixed models. LR test applied to a generalized lineal model (GLM) were conducted to compare the parasitism rates of SSB eggs by *T. japonicum* (binomial distribution error with logit link function). Two-way and one-way analysis of variance (ANOVA) followed by least significant difference (LSD) test were used to compare the body weight increases of the SSB

larvae on different plant treatments. The contents of JA, SA, and TPIs in different samples were analyzed using one-way ANOVA followed by Tukey's honest significant difference (HSD) test or two-sided Student's *t*-test. Behavioral responses of *T. japonicum* in Y-tube assays were analyzed using binomial test with an expected response of 50% for either olfactometer arm; parasitoids that did not make a choice were excluded from the analysis. Differences in volatile emission and in gene expression were analyzed by partial least squares-discriminant analysis (PLS-DA)[27,67] using SIMCA 14.1 software. The omics data were normalized by medians, log-transformed, and then auto-scaled (mean centered and divided by the standard deviation of each variable) using Metaboanalyst 4.0 software[83] before they were subjected to PLS-DA. The significance of treatment differences in PLS-DA was assessed using a permutation analysis (999 repetitions) implemented in the MVA.test from the RVAideMemoire package[84].

## Data availability

Data supporting the findings of this study are available within the paper and its Supplementary Information files. RNA-seq raw data have been deposited to the Gene Expression Omnibus (GEO) database in the National Center for Biotechnology Information (NCBI) with accession codes GSE167872. The sequences of rice genes were downloaded from the Phytozome database (https://phytozome-next.jgi.doe.gov/). Source data are provided with this paper.

## Code availability

The code used to conduct the analyses is available on request from the authors.

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

## Acknowledgements

We are grateful to Dr. Carlos Bustos-Segura at the University of Neuchâtel for helping with statistical analyses and Dr. Wei Liu (Institute of Plant Protection, CAAS) for assistance in making the insects illustrations. Dr. Thomas Degen provided valuable assistance with writing the manuscript. The study was supported by the National Natural Science Foundation of China (32120103009 and 31972984 to Y.Li). The contribution by T.C.J.T. was supported by European Research Council Advanced Grant 788949.

## Author contributions

Y.Li conceived and directed the project. Y.Li, Q.L., X.H. and T.C.J.T. designed the study. X.H. and S.S. performed the experiments. Y.N. provided the rice mutants of the bowman-birk inhibitor *AvrPiz-t interacting protein 4*. Q.L., X.H., S.S., Y.Li and T.C.J.T. analyzed the data. Q.L., X.H., S.S., Y.P., G.Y., Y.Lou, T.C.J.T. and Y.Li wrote the manuscript. All authors have read and approved the manuscript for publication.

## Competing interests

The authors declare no competing interests.
