## [Peer Review File · Nature Communications]

Reviewers' Comments:

Reviewer #1:

None

Reviewer #2:

Remarks to the Author:

Following on from one of their previous papers (Hu et al., eLife, 2020), in which the authors presented evidence that herbivory by the rice striped stem borer (SSB) *Chilo suppressalis* may facilitate rice infestation by the brown planthopper (BPH) *Nilaparvata lugens*, they present here exciting new data that the favor appears to be returned and that BPH feeding may reciprocally benefit SSB feeding as well. The authors show elegantly and thoroughly that this is the case for this particular combination of rice variety (the indica variety Minghui63), isolate of SSB, isolate of BPH, isolate of the SSB egg parasitoid *Trichogramma japonicum*, and isolate of the BPH egg parasitoid *Anagrus nilaparvatae* (the latter from the previous paper Hu et al., eLife, 2020).

While the ecological outcome of this set of interactions is fascinating and may have very important implications for our understanding of species interactions in rice agro-ecosystems and sustainable pest management therein, the authors make a leap to suggest that SSB and BPH may have co-evolved and now cooperate to colonize rice hosts. This is certainly possible, but the authors do not provide data to back up this suggestion. With the ecological data they provide for this combination of isolates from each species this is merely an evolutionary "just so story". Yet, it is this appearance of co-evolution that the whole manuscript revolves around and it is mentioned prominently in the abstract, introduction and discussion.

While I do not dispute that co-evolution is possible, if the authors wish to establish that there may indeed be co-evolution at play here, then it would be important to find out if the three conditions that Woolhouse et al. (Nature Genetics, 2002) list for coevolution to take place have been met here: (i) genetic variation in the relevant traits for all relevant species that are interacting here (e.g. plant resistance and tolerance, BPH and SSB feeding/oviposition); (ii) reciprocal effects of these relevant traits on fitness of all relevant species; (iii) dependence of the outcome of the species interactions on the combination of genotypes of the different species involved.

In the data that the authors present there is at best partial evidence that these conditions are met. Nonetheless, I feel like the authors' findings are an important contribution in the sense that this is one of the first mechanistic studies of how two different herbivorous insects appear to benefit from attacking shared host plants at or around the same time. A stronger emphasis on the ecological and pest management implications and, bar the addition of data in line with what Woolhouse et al. suggest, a toning down of the evolutionary claims would fit better with the data currently presented.

A second important point in my view is that the transcriptome and volatile emission data appear to be under-analyzed. The analyses of gene expression focus on regulation by jasmonic acid (JA) and salicylic acid (SA). Yet, it has been clear for years now that other plant hormones and signaling compounds may make important contributions too, including ethylene, gibberellins, etc. And the finding that dual infestation appears to cause down-regulation of both JA and SA levels in Fig 4 points to this as well. The HORMONOMETER analysis provides some insight into these contributions and it would be good if there could be some explanation of whether any hormones other than JA or SA may be important here. For the different sets of up- and down-regulated genes in each comparison it would be helpful to see if the promoters of these genes are enriched in binding sites for certain transcription factors (TFs). These TFs could fuel hypotheses for mechanistic links between hormonal regulation and gene expression changes.

And how do the gene expression data link up with the volatile emission data? Figures 4a and 5a show an interesting difference in the sense that in Fig 4a the "SSB" group stays with the "Control" group along PC2 while the "SSB/BPH" group is separated, whereas in Fig 5a the "SSB/BPH" group stays with the "Control" group along PC2 while the "SSB" group is separated. In addition, are there particular volatiles that appear to drive the separation between the different treatment groups?

Other points:

Results, line 221: It would be helpful to describe that the samples for RNA-seq were taken from plants after 48h of pre-treatment, so as SSB adults and larvae would have found such plants at the start of oviposition and body weight experiments such as presented in Figures 1 and 2.

Results, line 276: Is there any evidence in the literature that perhaps mutants in one or more proteinase inhibitor genes show differences in SSB resistance? This would add power to the suggestion that proteinase inhibitors could be important here.

This is the Woolhouse et al. paper that I mentioned earlier:

Woolhouse ME, Webster JP, Domingo E, Charlesworth B, Levin BR. Biological and biomedical implications of the co-evolution of pathogens and their hosts. *Nat Genet.* 2002 Dec;32(4):569-77. doi: 10.1038/ng1202-569. PMID: 12457190.

Reviewer #3:

Remarks to the Author:

This paper is a very clear, well-documented, solid and well-written study proving that two insect herbivores not only benefit from exploiting the same host plant, but show an adaptive behavior that makes them select the plants already infested by the other species. Facilitation for one of the two insect species was already published in a previous paper, while this new study documents that the benefit is reciprocal and finds it to be both direct and indirect. Additionally, this study brings interesting elements about the molecular mechanisms that may explain the reciprocal benefit of the co-infestation. Authors claim that this is the first evidence that two insect herbivores benefit from attacking the same individual plants. I am convinced with the results presented, which were obtained with well-designed and well-documented experiments and correct statistical analyses (although I have minor comments, see below).

I have no major concern about this study. All comments are rather minor, although some deserve careful consideration.

- Introduction:

* It is confusing that you speak of competition and mutual benefit only, which you seem to oppose in a binary scheme, while citing as first reference a paper from 2007 concluding that "Clearly, a new paradigm that accounts for indirect interactions and facilitation is required to describe how interspecific competition contributes to the organization of phytophagous insect communities." I would suggest to introduce facilitation more clearly and to discuss the relevance of mutual benefit relative to facilitation.

* Please avoid terms like "higher organisms".

- Materials and Methods (statistical analyses):

* Oviposition tests of SSB females: How did you account for overdispersion with a Poisson distribution? Additionally, please say clearly what the random factor is.

* Parasitism rates: The arcsin square-root transformation is old-fashion and not really justifiable since GLMs with binomial distributions are easily applicable. I strongly suggest using such GLMs for these data.

* Omics data: How could you apply a log transformation to VOCs data that contain many zeroes? Additionally, you cannot conclude anything based only on the graph of a PLS-DA since overfitting can lead to erroneous conclusions (see Westerhuis et al. (2008) Assessment of PLS-DA cross validation. *Metabolomics*, 4: 81-89). You must perform a significance test before drawing any conclusion from a PLS-DA. Such test is implemented e.g. in the `MVA.test()` function of the `RVAideMemoire` package in R.

- Results:

* SSB performance: Please clarify the last sentence that is quite confusing.

* Oviposition tests of SSB females: Here and everywhere else, replace "RT-test" with "LR-test" and add degrees of freedom.

* PLS-DAs: replace "principal components" with "components" since it introduces confusion with PCA.

* Volatile profiles: Where do the results on amounts of VOCs emitted come from? There is nothing about that in the M&M section. Additionally, if these total amounts were calculated using % areas relative to the internal standard, as I suspect, it is wrong to sum them since all compounds do not have the same response factor.

* Oviposition tests of the parasitoid: I do not agree that the results show a repellence of BPH-infested plants. This might be true, but it may also be that masking compounds are emitted, or that the volatile message is changed and not recognized anymore.

- Discussion:

* I am not sure you can say that both herbivores benefit directly from co-infestation since it leads to mitigated plant toxicity. Your results suggest that this is true for SSB, but if I am correct, you did not show that for BPH in your previous study.

* I would remove the sentences about the differential feeding strategies, which are too speculative and do not bring anything to the paper.

- In general:

* Please check all references to figures as several are not correct.

Response to Referees Letter

Dear reviewers,

Thanks a lot for the largely positive comments made by you on our study. We are impressed by the careful and detail review of the paper and thankful for the constructive comments and suggestions, which were very useful to improve the manuscript. Based on comments made by you we conducted additional key experiments and have made significant changes to the manuscript.

Below we describe in detail how we have addressed each comment. Two versions of the revised manuscript have been submitted, one with track-changes to show what was changed, and one clean version for the next step of the review process.

We hope that you are satisfied with the revision and explanations given below. We would be happy to take any other suggestions for the improvement of the manuscript into account and we look forward to further feedback.

Kind regards,

Yunhe Li & Ted Turlings

Note: The line numbers provided in the following texts are the line numbers in the manuscript file with tracked changes.

REVIEWER COMMENTS

Reviewer #2 (Remarks to the Author):

Following on from one of their previous papers (Hu et al., eLife, 2020), in which the authors presented evidence that herbivory by the rice striped stem borer (SSB) *Chilo suppressalis* may facilitate rice infestation by the brown planthopper (BPH) *Nilaparvata lugens*, they present here exciting new data that the favor appears to be returned and that BPH feeding may reciprocally benefit SSB feeding as well. The authors show elegantly and thoroughly that this is the case for this particular combination of rice variety (the indica variety Minghui63), isolate of SSB, isolate of BPH, isolate of the SSB egg parasitoid *Trichogramma japonicum*, and isolate of the BPH egg parasitoid *Anagrus nilaparvatae* (the latter from the previous paper Hu et al., eLife, 2020).

While the ecological outcome of this set of interactions is fascinating and may have very important implications for our understanding of species interactions in rice agro-ecosystems and sustainable pest management therein, the authors make a leap to suggest that SSB and BPH may have co-evolved and now cooperate to colonize rice hosts. This is certainly possible, but the authors do not provide data to back up this suggestion. With the ecological data they provide for this combination of isolates from each species this is merely an evolutionary “just so story”. Yet, it is this appearance of co-evolution that the whole manuscript revolves around and it is mentioned prominently in the abstract, introduction and discussion.

While I do not dispute that co-evolution is possible, if the authors wish to establish that there may indeed be co-evolution at play here, then it would be important to find out if the three conditions that Woolhouse et al. (Nature Genetics, 2002) list for coevolution to take place have been met here: (i) genetic variation in the relevant traits for all relevant species that are interacting here (e.g. plant resistance and tolerance, BPH and SSB feeding/oviposition); (ii) reciprocal effects of these relevant traits on fitness of all relevant species; (iii) dependence of the outcome of the species interactions on the combination of genotypes of the different species involved.

In the data that the authors present there is at best partial evidence that these conditions are met. Nonetheless, I feel like the authors' findings are an important contribution in the sense that this is one of the first mechanistic studies of how two different herbivorous insects appear to benefit from attacking shared host plants at or around the same time. A stronger emphasis on the ecological and pest management implications and, bar the addition of data in line with what Woolhouse et al. suggest, a toning down of the evolutionary claims would fit better with the data currently presented.

R: We agree with virtually all of these comments and the reviewer is right that more concrete evidence for a truly co-evolutionary process would be pertinent. Yet, we think, and hope that the reviewer agrees, that based on commonly used definitions, we are correct to conclude that the here revealed plant-mediated interaction between the rice herbivores constitutes cooperation, which to our knowledge has not previously been reported. We also agree with the reviewer that the system of SSB and BPH on rice provides an ideal model to conclusively show that the interaction is the result of co-evolution. In fact, we already have an additional data set with another moth, the rice leaf folder (RLF), that confirms the uniqueness of the SSB-BPH (see the figure below). We could include this, but this would make the paper harder to follow for readers and we prefer to keep this for a follow up study that we just started. In this new study, we aim to confirm that reciprocal selection (co-evolution) has favored the cooperative behavior of sympatric SSB and BPH by including populations where SSB and BPH do not co-occur. We have already established BPH colonies that originate from Xinjiang and Jilin provinces, where there are virtually no SSB. We will conduct

similar experiments with these colonies and those where the insects live in sympatry. If the results fit our hypothesis, we are close to fulfilling the conditions listed by Woolhouse et al. 2002 in favor of co-evolution. It will take at least 3 years to finish the experiments that we have in mind and they should therefore logically be part of a future publication.

In brief, we are delighted that the reviewer recognizes that we present “(one of) the first mechanistic studies of how two different herbivorous insects appear to benefit from attacking shared host plants”, but we agree with the reviewer that we cannot yet conclude that this the result of co-evolution. This does not take away from our main conclusions. We have adapted the text accordingly and hope that reviewer agrees that the conclusions and uniqueness of our study remain fully valid.

Fig.1 Weight of the rice leaf folder (RLF) and the brown planthopper (BPH) after seven days of feeding on differently infested rice plants. Neonates of RLF (a) and nymphs of BPH (b) were introduced on rice plants that were either uninfested (Control), infested by RLF larvae only (RLF), BPH only (BPH), or both RLF and BPH (RLF/BPH, two species were simultaneously introduced to the plants). These

performance results show that BPH and RLF are competitors when feeding on the same plant. This confirms the uniqueness of the cooperative interaction between BPH and SSB.

A second important point in my view is that the transcriptome and volatile emission data appear to be under-analyzed. The analyses of gene expression focus on regulation by jasmonic acid (JA) and salicylic acid (SA). Yet, it has been clear for years now that other plant hormones and signaling compounds may make important contributions too, including ethylene, gibberellins, etc. And the finding that dual infestation appears to cause down-regulation of both JA and SA levels in Fig 4 points to this as well. The HORMONOMETER analysis provides some insight into these contributions and it would be good if there could be some explanation of whether any hormones other than JA or SA may be important here. For the different sets of up- and down-regulated genes in each comparison it would be helpful to see if the promoters of these genes are enriched in binding sites for certain transcription factors (TFs). These TFs could fuel hypotheses for mechanistic links between hormonal regulation and gene expression changes.

R: This is also a valid comment. We have now included additional analyses of other plant hormones such as ethylene, auxin, gibberellins, abscisic acid, cytokinin, brassinosteroid using our transcriptome data (see supplementary Fig. 1). These analyses did not reveal any consistent correlations between these other hormones and the production of protease inhibitors, which are known to be the main defense metabolites against lepidopteran pest including SSB (*Farmer et al., 1992. Plant Physiology, 98, 995-1002; Koiwa et al., 1997. Trends in Plant Science, 2, 379-384*). In addition, the correlation between protease inhibitors and SSB-resistance were further confirmed with additional experiments using transgenic rice lines in which the trypsin protease inhibitors gene (APIP4: Os01g0124200) was knocked out or overexpressed (see Fig. 5). Furthermore, the interaction between the defense hormone JA and protease inhibitor production has been well established in the previous studies by the group of co-author Prof. Yonggen Lou and other experts (*Zhou et al., 2009. The Plant Journal, 60, 638-648; Lu et al., 2014. Molecular Plant, 7, 1670-1682; Zeng et al., 2021. Plants, 10, 442*) and other studies (*Ye et al., 2013. PNAS, 110(38): E3631-E3639; Ye et al., 2012. PLoS ONE, 7(4): e36214; Ogawa et al., 2017. Biochemical and Biophysical Research Communications, 486, 796-803*). Moreover, cross-talk between SA has been reported several times (*Guo et al., 2018. Current Opinion in Plant Biology, 44:72-81; Erb, M., & Reymond, P. 2019. Annual Review of Plant Biology, 70, 527-557; Aerts et al., 2021. The Plant Journal, 105, 489-504*). We therefore made the decision to focus on the analysis of these two major plant defense hormones. In the revised version, it is now explained why JA and SA were selected for detailed analyses (**Lines 267-271 and Lines 757-760**) and a supplementary Fig. 1 was added to show the gene expression of rice hormones other than JA and SA, following different pest infestations.

As suggested by the reviewer, we further conducted transcription factors (TFs) and

promoter analyses. Gene IDs of TFs in rice were retrieved from platform PlantRegMap and were then compared to the downregulated DEGs between dual infestation and SSB infestation treatments to identify potential regulated TFs. To identify upstream regulatory TFs, the 2000-bp sequences upstream of the start codon of the JA signaling genes were extracted from the Phytozome database (<https://phytozome-next.jgi.doe.gov/>). Then, these sequences were subjected to the Plant RegMap with a threshold P-value $\leq 1e-5$ for TF binding site prediction. The motifs of TFs that have the potential to bind to the promoters of the JA signaling genes were screened, and a possible interaction was assigned if there were one or more binding sites of a TF on the promoter of a JA signaling gene. The predicted TFs were matched with the identified TFs that were downregulated in dual infestation as compared to SSB infestation, in order to obtain the potentially co-expressed TFs. The results show that four TFs, *OsNAC39* (*Os03g0327100*), *OsIDD1* (C2H2 type family protein) (*Os03g0197800*), *OsHOX21* (HD-ZIP family protein) (*Os03g0170600*), and *OsSta2* (ERF family protein) (*Os02g0655200*), are predicted to bind to promoters of genes associated with JA pathway (see supplementary data5). Moreover, the regulation of these TFs and JA pathway genes were found to be similar (supplementary data5). These results suggest that the identified TFs may control the expression of JA pathways genes. We have added related information in the methods, results and discussion parts in the revised version (**Lines 280-289, Lines 513-520, and Lines 776-794**). This additional information should help readers to better understand the role of the JA pathway in the rice-mediated insect interactions described in the current study.

And how do the gene expression data link up with the volatile emission data? Figures 4a and 5a show an interesting difference in the sense that in Fig 4a the “SSB” group stays with the “Control” group along PC2 while the “SSB/BPH” group is separated, whereas in Fig 5a the “SSB/BPH” group stays with the “Control” group along PC2 while the “SSB” group is separated. In addition, are there particular volatiles that appear to drive the separation between the different treatment groups?

R: This is again a good comment. In fact, we did try to establish the link between gene expression and the volatile emission data. Based on our current study (Supplementary Data 6) and the data from our previous paper (Hu et al., 2020. eLife, 9, e55421), many volatiles are released in response to infestation by the different insects. There are no obvious differences in the identity of the volatile compounds, but the differences between treatment groups are rather the consequence of changes in relative ratios of the complex volatile blends. We could not find particular volatiles driving the separation between the treatments. And only few genes coding for volatile rice compounds are currently known, it is therefore not yet impossible to build links between genes and volatile emissions.

In addition, we should also note that the rice plant samples for transcriptome analysis were collected during daytime, while the volatiles that affect moth behavior are emitted during nighttime. It is therefore not appropriate to link our current gene

expression data with the volatile emission data. Although we believe that such information would be valuable, it seems not really necessary for the main objective of the current study. Therefore, we prefer to leave such experiments and analyses for our future study.

Other points:

Results, line 221: It would be helpful to describe that the samples for RNA-seq were taken from plants after 48h of pre-treatment, so as SSB adults and larvae would have found such plants at the start of oviposition and body weight experiments such as presented in Figures 1 and 2.

R: Good suggestion. We now emphasized this point of 48 hr of pre-treatment before sampling for RNA-seq (**Line 233**).

Results, line 276: Is there any evidence in the literature that perhaps mutants in one or more proteinase inhibitor genes show differences in SSB resistance? This would add power to the suggestion that proteinase inhibitors could be important here.

R: This is a very good suggestion.

Although there is indirect evidence showing the resistance of trypsin protease inhibitors against SSB, no direct evidence was available that shows that proteinase inhibitors in rice plants confer resistance to SSB caterpillars. We now provide such evidence.

We were kindly provided with transgenic rice lines with the knockout or overexpression of the trypsin protease inhibitors gene APIP4 (ID: Os01g0124200) by our colleague Prof. Yuese Ning (for details to see Zhang et al., 2020. *Plant Biotechnology Journal*, 18, 2354-2363). Our transcriptomic analysis suggests that this trypsin protease inhibitors gene plays an important role in rice defensive response to SSB infestation (see Fig.4f). With the transgenic rice lines, we conducted an additional bioassay (**Lines 822-840**) to fully confirm this importance of trypsin protease inhibitors in providing resistance of against SSB (see Fig. 5). These additional bioassays further support our conclusions. We thank the reviewer for this valuable suggestion.

This is the Woolhouse et al. paper that I mentioned earlier:

Woolhouse ME, Webster JP, Domingo E, Charlesworth B, Levin BR. Biological and biomedical implications of the co-evolution of pathogens and their hosts. *Nat Genet.* 2002 Dec;32(4):569-77. doi: 10.1038/ng1202-569. PMID: 12457190.

R: We now cite this valuable paper in the revised version of our manuscript (**Line 47**).

Reviewer #3 (Remarks to the Author):

This paper is a very clear, well-documented, solid and well-written study proving that

two insect herbivores not only benefit from exploiting the same host plant, but show an adaptive behavior that makes them select the plants already infested by the other species. Facilitation for one of the two insect species was already published in a previous paper, while this new study documents that the benefit is reciprocal and finds it to be both direct and indirect. Additionally, this study brings interesting elements about the molecular mechanisms that may explain the reciprocal benefit of the co-infestation. Authors claim that this is the first evidence that two insect herbivores benefit from attacking the same individual plants. I am convinced with the results presented, which were obtained with well-designed and well-documented experiments and correct statistical analyses (although I have minor comments, see below).

R: We thank the reviewer for the positive comments.

I have no major concern about this study. All comments are rather minor, although some deserve careful consideration.

- Introduction:

* It is confusing that you speak of competition and mutual benefit only, which you seem to oppose in a binary scheme, while citing as first reference a paper from 2007 concluding that “Clearly, a new paradigm that accounts for indirect interactions and facilitation is required to describe how interspecific competition contributes to the organization of phytophagous insect communities.” I would suggest to introduce facilitation more clearly and to discuss the relevance of mutual benefit relative to facilitation.

R: We have revised the introduction in accordance with these remarks and now introduce the concept of “facilitation”, adding several relevant references (**Lines 73-79, Lines 88-92, and Lines 118-122**).

* Please avoid terms like “higher organisms”.

R: We have changed the term “higher organisms” to “vertebrates” (**Line 50**).

- Materials and Methods (statistical analyses):

* Oviposition tests of SSB females: How did you account for overdispersion with a Poisson distribution? Additionally, please say clearly what the random factor is.

R: We used cage ID as a random factor to account for the non-independence between observations within the same cage. Whenever the dispersion value was higher than 2 in Poisson GLMMs, overdispersion was accounted for by using an observation level factor as a random factor. Thus, cage ID and an observation level factor were the two random factors in the mixed models. This information has been added to the description of the statistical analyses in the revised version (**Lines 965-970**).

* Parasitism rates: The arcsin square-root transformation is old-fashion and not really justifiable since GLMs with binomial distributions are easily applicable. I strongly suggest using such GLMs for these data.

R: We thank the reviewer for this suggestion. We re-analyzed the data of parasitism rates directly using Generalized Linear Models (GLM) with binomial distribution error followed by least significance difference post hoc tests (**Lines 970-974**). With the new statistical results, we found that the parasitism rates of SSB eggs even better reflect the responses of the parasitoids to treated plants (Fig. 7), for example we detected a statistical difference between uninfested plants and plants infested by SSB larvae, while no significantly difference was detected with the “transformed data”. Thanks again for the suggestion.

* Omics data: How could you apply a log transformation to VOCs data that contain many zeroes? Additionally, you cannot conclude anything based only on the graph of a PLS-DA since overfitting can lead to erroneous conclusions (see Westerhuis et al. (2008) Assessment of PLS-DA cross validation. *Metabolomics*, 4: 81-89). You must perform a significance test before drawing any conclusion from a PLS-DA. Such test is implemented e.g. in the `MVA.test()` function of the `RVAideMemoire` package in R.

R: We analyzed the omics data using Metaboanalyst 4.0 software with the default parameter. Specifically, the data were first normalized by median, and then were subjected to log transformation and auto scaling (mean-centered and divided by the standard deviation of each variable). The data that contained zeroes were first normalized by median, and the obtained new datasets contained no zeroes before they were processed with log transformation and auto scaling.

For the PLS-DA, we agree with the reviewer that overfitting can lead to erroneous conclusions. As suggested by the reviewer, the significance of the treatment was assessed using a permutation analysis (999 repetitions) implemented in the `MVA.test` from the `RVAideMemoire` package (**Lines 988-991**). The results show that there were significant differences among the four treatments in both VOCs data (classification error = 19.1%, $P = 0.001$) (**Lines 379-382**) and gene expression (classification error = 44.2%, $P = 0.04$) (**Lines 236-237**). Therefore, we believe that PLS-DA were appropriately used for our omics data. To be clearer, we added the statistical results in the revised version. Thanks for the suggestion.

- Results:

* SSB performance: Please clarify the last sentence that is quite confusing.

* Oviposition tests of SSB females: Here and everywhere else, replace “RT-test” with “LR-test” and add degrees of freedom.

R: We apologize for the confusion. We have revised this sentence to make it clearer.

In addition, we have modified the text “RT-test” into “LR-test” accordingly and added more information about the statistics, including the degrees of freedom and chi-square values (**Lines 163-185, Lines 208-216, and Lines 392-394**).

* PLS-DAs: replace “principal components” with “components” since it introduces confusion with PCA.

R: Thanks for the suggestion. We modified the text as suggested (**Lines 233-241**).

* Volatile profiles: Where do the results on amounts of VOCs emitted come from? There is nothing about that in the M&M section. Additionally, if these total amounts were calculated using % areas relative to the internal standard, as I suspect, it is wrong to sum them since all compounds do not have the same response factor.

R: In the previous version, we did not provide details regarding collection and identification of VOCs, but only cite our previous paper using the same method (Hu et al., 2020, eLife). Because of this comment from the reviewer, we re-organized the method section and added necessary details (**Lines 846-884**).

For the quantification of the volatiles, we do agree that the calculation using % areas relative to the internal standard cannot provide precise quantifications because different compounds have different response factors. Such data are not appropriate for calculation of the total amounts of volatiles by summing individual ones as pointed out by the reviewer.

However, for the current study we were not interested in the total amounts of volatiles emitted from plants, we were rather interested in the differences between plant treatments for each individual compound. For this purpose, the calculation of relative quantification of the volatiles is valid and suitable.

* Oviposition tests of the parasitoid: I do not agree that the results show a repellence of BPH-infested plants. This might be true, but it may also be that masking compounds are emitted, or that the volatile message is changed and not recognized anymore.

R: We agree with this concern. We have revised the sentence (**Lines 435-437**) into “These results imply that additional infestation by BPH changes rice volatiles so that *T. japonicum* wasps no longer recognize and locate the rice plants carrying SSB-eggs.”

- Discussion:

* I am not sure you can say that both herbivores benefit directly from co-infestation since it leads to mitigated plant toxicity. Your results suggest that this is true for SSB, but if I am correct, you did not show that for BPH in your previous study.

R: The reviewer is right, the evidence for BPH was solely based on an ambiguous

interpretation of our previous finding that SSB-infested rice plants appeared more suitable for BPH performance (Wang et al., 2018. Plant Biotech J. 16: 1748-1755).

Because of the reviewer's comment we conducted an additional bioassay that confirms that BPH performs significantly better on SSB-infested rice than on uninfested rice plants (see Supplementary Fig.3). We therefore can safely conclude that both herbivores *including BPH* benefit directly from co-infestation.

In the new version of the manuscript, the description of our previous and new findings regarding performance of BPH feeding on SSB-infested rice plants was further improved in the introduction (**Lines 87-93**) and discussion (**Lines 479-480**).

* I would remove the sentences about the differential feeding strategies, which are too speculative and do not bring anything to the paper.

R: We agree, and the related sentences were deleted (**Lines 572-580**).

- In general:

* Please check all references to figures as several are not correct.

R: Thank you for your careful review. We have revised the references to figures to correctly assign them to the corresponding figures.

Reviewers' Comments:

Reviewer #2:

Remarks to the Author:

The authors have gone above and beyond with addressing my comments on the first version of the manuscript. I have no further qualms and would like to congratulate the authors on putting together such an impressive synthesis.

There is a minor typo that I think I spotted:

In line 263:

"As expected, TPI content was significantly higher in apip4-5 plants ($P < 0.05$) and significantly lower in APIP4-OX 16-2 lines ($P < 0.01$) relative to WT plants."

Should "higher" and "lower" be the other way around?

Reviewer #3:

Remarks to the Author:

I thank the authors for their careful consideration of my previous comments.